# Variational excess risk bound for general state space models

**Elisabeth Gassiat**  *elisabeth.gassiat@universite-paris-saclay.fr*
*Université Paris-Saclay, CNRS*
*Laboratoire de mathématiques d'Orsay*
*91405, Orsay, France*

**Sylvain Le Corff**  *sylvain.le__corff@sorbonne-universite.fr*
*LPSM*
*Sorbonne Université, UMR CNRS 8001*
*Paris, France*

**Reviewed on OpenReview:** *https://openreview.net/forum?id=36OX7uRM5t*

## Abstract

In this paper, we consider variational autoencoders (VAE) for general state space models. We consider a backward factorization of the variational distributions to analyze the excess risk associated with VAE. Such backward factorizations were recently proposed to perform online variational learning Campbell et al. (2021) and to obtain upper bounds on the variational estimation error Chagneux et al. (2022). When independent trajectories of sequences are observed and under strong mixing assumptions on the state space model and on the variational distribution, we provide an oracle inequality explicit in the number of samples and in the length of the observation sequences. We then derive consequences of this theoretical result. In particular, when the data distribution is given by a state space model, we provide an upper bound for the Kullback-Leibler divergence between the data distribution and its estimator and between the variational posterior and the estimated state space posterior distributions. Under classical assumptions, we prove that our results can be applied to Gaussian backward kernels built with dense and recurrent neural networks.

## 1 Introduction

Deep generative models have been increasingly used and analyzed for the past few years. In this setting, Variational autoencoders (VAEs) offer the possibility to simultaneously model and train (i) the conditional distribution of the observation given latent variables referred to as the decoder, and (ii) a variational approximation of the conditional distribution of the latent variable given the observation referred to as the encoder. They have been successfully applied in many contexts such as image generation (Vahdat & Kautz, 2020), text generation (Bowman et al., 2015), state estimation and image reconstruction (Cohen et al., 2022).

Variational inference has been widely and satisfactorily used for many practical applications but its theoretical properties has been analyzed only very recently. Theoretical guarantees have been mostly proposed for variational inference procedures in settings where datasets are based on independent data and for mean-field approximations. In Huggins et al. (2020), the authors provided variational error bounds, in particular for the estimation of the posterior mean and covariance. In Chérief-Abdellatif & Alquier (2018), the authors established the concentration of variational approximations of posterior distributions for mixtures of general laws using PAC-Bayesian theory. The PAC-Bayesian theory has also been used in Mbacke et al. (2023) where the authors controlled in particular the $L^2$ reconstruction loss under the true data distribution for VAEs. In addition, Tang & Yang (2021) provided a theoretical analysis of the excess risk for Empirical Bayes Variational Autoencoders for both parametric and nonparametric settings. They derived a set of generic assumptions to obtain an oracle inequality explicit in the number of samples and proposed an upper bound for the total variation distance between the true distribution of the observations and a variational approximation combining the empirical distribution of the dataset and the proposed VAE architecture.

In this paper, we consider data sets consisting in $n$ independent copies of sequences with length $T+1$ having distribution $P_{\mathcal{D}}$. We aim at extending the theoretical results on variational inference procedures in two directions. First, we set the focus on the use of VAEs for general state space models, i.e. settings where the decoding distribution $P_\theta^Y$ of the observations depends on an unobserved Markov chain. State space models are a ubiquitous class of latent variable models for sequential data, see for instance Marino et al. (2018); Lin et al. (2018); Krishnan et al. (2017). In addition, instead of using mean-field approximations, we consider variational encoding distributions $Q_\varphi$ satisfying a backward factorization as proposed in Campbell et al. (2021); Chagneux et al. (2022). In Chagneux et al. (2022), the authors derived the first theoretical results providing upper bounds on the state decoding estimation error when using variational inference with backward factorization and no such results were proposed for state space models using a mean-field approximation. This factorization was used in Campbell et al. (2021) to define new online variational estimation algorithms, where observations are processed on-the-fly. In state space models, the true posterior distribution of the latent states given the observations admits a backward Markovian decomposition. Therefore this factorization allows to introduce a variational family which fits the data structure which is not the case of mean-field approximations.

Our results give the first (up to our knowledge) theoretical guarantees on the trained variational approximation in this setting.

- We provide assumptions on the decoding and variational encoding kernels under which we prove an oracle inequality for the risk explicit in particular in the number of samples and in the length of the observation sequences, see Theorem 3.1. This result is established using an alternative formulation of (Tang & Yang, 2021, Theorem 3) in our state space setting and with an explicit dependency on some constants to track all terms depending on the number of observations. The variance term has the usual rate $1/n$ up to $\log n$ terms in the sample size $n$, and grows as $T^3$ in the length $T$ of the sample sequences. This allows to understand when the procedure leads to a decoding distribution that approximates well the data distribution together with a coding distribution which approximates well the decoding state distribution.

- In particular, when data are generated from a general state space model, and when $P_{\mathcal{D}}$ belongs to the decoding family of distributions, we give an upper bound also explicit in the way the backward coding kernels approximate the backward decoding kernels, see Corollary 1.

- We analyse settings in which our results hold, in particular settings with Gaussian backward kernels based on Multi-Layer Perceptrons (MLPs) and on Recurrent Neural Networks (RNNs).

Our theoretical results provide the first excess risk bounds in a context of VAE for state space models. The proposed upper bound has the same behaviour in $n$ as the ones obtained in the classical statistical literature for parametric models, in which $1/n$ is the parametric rate and $\log n$ factors come from the concentration of empirical measures. This behaviour was recovered in (Tang & Yang, 2021, Theorem 3) in the variational learning context. Regarding theoretical results for VAEs in the state space models context, we are only aware of Chagneux et al. (2022) where the authors control the variational posterior error for the estimation of expectations of additive functionals with an upper bound having linear growth in $T$. In this context, we target a more challenging objective and it is not surprising that the upper bound on the excess risk may have a larger than linear growth in $T$.

The paper is organised as follows. The general setting and notations for state space models and variational learning are given in Section 2. Assumptions and theoretical results are proposed in Section 3 along with discussions on specific deep architectures used in practice. A discussion with insights for future works is given in Section 4. Detailed technical setting ot the assumptions and detailed proofs of theoretical results are given in Appendices. Additional proofs to highlight that when the state and observation spaces are compact our main results hold are given in Appendix G.

## 2 State space model and variational estimation

The observations are $n$ sequences $Y_{0:T}^i$, $1 \leq i \leq n$, taking values in a measurable space $(\mathsf{Y}, \mathcal{Y})$. We use $a_{u:v}$ as a short-hand notation for $(a_u, \ldots, a_v)$ for $0 \leqslant u \leqslant v$ and any sequence $(a_\ell)_{\ell \geqslant 0}$. In the following, we need to consider quantities depending on observation sequences $y_{0:T}$, which is highlighted with super-indices in all what follows.

### 2.1 Coding distribution with state space modeling

Throughout the paper, all quantities related to the coding distribution depend on a parameter $\theta \in \Theta$, $\Theta \subset \mathbb{R}^{d_\theta}$. To be used as coding distribution for the observations, we consider a state-space model, i.e. a bivariate discrete-time process $\{(X_t, Y_t)\}_{t \geq 0}$ where $\{X_t\}_{t \geq 0}$ is a hidden Markov chain in a measurable space $(\mathsf{X}, \mathcal{X})$. The initial distribution $\chi$ of $X_0$ has density $\zeta$ with respect to a reference measure $\mu$ and for all $t \geqslant 0$, the conditional distribution of $X_{t+1}$ given $X_{0:t}$ depends only on $X_t$, it is written $M_\theta(X_t, \cdot)$ and has density $m_\theta(X_t, \cdot)$. In such a model, the observations $\{Y_t\}_{0 \leqslant t \leqslant T}$ are assumed to be independent conditionally on $X_{0:T}$ and, for all $0 \leqslant t \leqslant T$, the distribution of $Y_t$ given $X_{0:T}$ depends on $X_t$ only, is written $G_\theta(X_t, \cdot)$, and has density $y \mapsto g_\theta^y(X_t)$ with respect to a reference measure $\nu$.

**Example 1.** *Consider a nonlinear state space model defined by $X_0 \sim \zeta$ in $\mathbb{R}^d$ and for all $0 \leq t \leq T - 1$,*

$$X_{t+1} = f_\theta(X_t) + \varepsilon_t \,,$$

*where $f_\theta : \mathbb{R}^d \to \mathbb{R}^d$ is a parametric function and $(\varepsilon_t)_{1 \leq t \leq T}$ are i.i.d. with distribution $\mathcal{N}(0, \Sigma)$ and independent of $X_0$. In this setting, $m_\theta(X_t, \cdot)$ is the Gaussian density with mean $f_\theta(X_t)$ and variance $\Sigma$. The observations are defined, for all $0 \leq t \leq T$, as*

$$Y_t = h_\theta(X_t) + \eta_t \,,$$

*where $h_\theta : \mathbb{R}^d \to \mathbb{R}^q$ is a parametric function and $(\eta_t)_{0 \leq t \leq T}$ are i.i.d. with distribution $\mathcal{N}(0, \Lambda)$ and independent of $(X_{0:T}, \varepsilon_{0:T})$. In this setting, $y \mapsto g_\theta^y(X_t)$ is the Gaussian density with mean $h_\theta(X_t)$ and variance $\Lambda$.*

In this context, the joint probability distribution $P_\theta$ of $(X_{0:T}, Y_{0:T})$ has density with respect to $\mu^{\otimes(T+1)} \otimes \nu^{\otimes(T+1)}$ given, for all $\theta \in \Theta$, $x_{0:T} \in \mathsf{X}^{T+1}$ and all $y_{0:T} \in \mathsf{Y}^{T+1}$, by

$$p_{\theta,0:T}(x_{0:T}, y_{0:T}) = \zeta(x_0) g_\theta^{y_0}(x_0) \prod_{t=1}^{T} m_\theta(x_{t-1}, x_t) g_\theta^{y_t}(x_t) \,,$$

and the coding distribution of a sequence $Y_{0:T}$ is denoted by $P_\theta^Y$. The joint smoothing distribution, i.e. the conditional distribution of $X_{0:T}$ given $Y_{0:T} = y_{0:T}$, is given for every measurable function $h$ by

$$\Phi_{\theta,0:T|T}^{y_{0:T}}(h) = \frac{\int \chi(\mathrm{d}x_0) g_\theta^{y_0}(x_0) \prod_{t=1}^{T} M_\theta(x_{t-1}, \mathrm{d}x_t) g_\theta^{y_t}(x_t) h(x_{0:T})}{\int \chi(\mathrm{d}x_0) g_\theta^{y_0}(x_0) \prod_{t=1}^{T} M_\theta(x_{t-1}, \mathrm{d}x_t) g_\theta^{y_t}(x_t)}\,.$$

The probability density of $\Phi_{\theta,0:T|T}^{y_{0:T}}$ is denoted by $\phi_{\theta,0:T|T}^{y_{0:T}}$. In the following, we use the notation $\Phi_{\theta,t}^{y_{0:t}}$ to denote the the filtering distribution at time $t$, i.e. the conditional distribution of $X_t$ given $Y_{0:t} = y_{0:t}$, with a similar convention for the probability densities. The joint smoothing distribution can also be written

$$\Phi_{\theta,0:T|T}^{y_{0:T}}(\mathrm{d}x_{0:T}) = \Phi_{\theta,T}^{y_{0:T}}(\mathrm{d}x_T) \prod_{t=0}^{T-1} B_{\theta,T-t-1|T-t}^{y_{0:T-t-1}}(x_{T-t}, \mathrm{d}x_{T-t-1}) \,, \tag{1}$$

where $B_{\theta,T-t-1|T-t}^{y_{0:T-t-1}}(x_{T-t}, \mathrm{d}x_{T-t-1})$ is the backward kernel at time $T-t$ defined by $B_{\theta,T-t-1|t}^{y_{0:T-t-1}}(x_{T-t}, \mathrm{d}x_{T-t-1}) \propto \Phi_{\theta,T-t-1}^{y_{0:T-t-1}}(\mathrm{d}x_{T-t-1}) m_\theta(x_{T-t-1}, x_{T-t})$ with a probability density with respect to $\mu$ denoted by $b_{\theta,T-t-1|T-t}^{y_{0:T-t-1}}(x_{T-t}, \cdot)$. For all $T$, $\theta$, $y_{0:T} \in \mathsf{Y}^{T+1}$, the loglikelihood of the observations is:

$$\ell_T^{y_{0:T}}(\theta) = \log L_T^{y_{0:T}}(\theta) \,,$$

where

$$L_T^{y_{0:T}}(\theta) = \int p_{\theta,0:T}(x_{0:T}, y_{0:T}) \mu(\mathrm{d}x_{0:T}) \,.$$

## 2.2 Variational learning with backward factorization

The joint smoothing distribution is usually intractable and we focus in this paper on variational learning to perform approximate maximum likelihood. All quantities related to the encoding distribution depend on a parameter $\varphi \in \Phi$, $\Phi \subset \mathbb{R}^{d_\varphi}$. Following Campbell et al. (2021); Chagneux et al. (2022), we propose a backward variational formulation mimicking (1:

$$Q_{\varphi,0:T}^{y_{0:T}}(\mathrm{d}x_{0:T}) = Q_{\varphi,T}^{y_{0:T}}(\mathrm{d}x_T) \prod_{t=0}^{T-1} Q_{\varphi,T-t-1|T-t}^{y_{0:T}}(x_{T-t}, \mathrm{d}x_{T-t-1}),$$

where $Q_{\varphi,T-t-1|T-t}^{y_{0:T}}(x_{T-t}, \cdot)$ (resp. $Q_{\varphi,T}^{y_{0:T}}$) has probability density $q_{\varphi,T-t-1|T-t}^{y_{0:T}}(x_{T-t}, \cdot)$ (resp. $q_{\varphi,T}^{y_{0:T}}$) with respect to the reference measure $\mu$.

In this setting, the ELBO writes, for all $\theta \in \Theta$, $\varphi \in \Phi$, and for a sequence of observations $Y_{0:T}$,

$$\mathrm{ELBO}_T^{Y_{0:T}}(\theta, \varphi) = \ell_T^{Y_{0:T}}(\theta) - \mathrm{KL}\left(Q_{\varphi,0:T}^{Y_{0:T}} \middle\| \Phi_{\theta,0:T|T}^{Y_{0:T}}\right).$$

Here, $\mathrm{KL}(Q\|P)$ denotes the Kullback-Leibler divergence between probability distributions $Q$ and $P$, that is $\mathrm{KL}(Q\|P) = \mathbb{E}_Q[\log(\mathrm{d}Q/\mathrm{d}P)]$. Maximizing the empirical ELBO given by $\sum_{i=1}^n \mathrm{ELBO}_T^{Y_{0:T}^i}(\theta, \varphi)$ is equivalent to minimizing the following loss function

$$\mathcal{L}_{n,T}(\theta, \varphi) = \frac{1}{n} \sum_{i=1}^n \varpi(\theta, \varphi, Y_{0:T}^i),$$

where

$$\varpi(\theta, \varphi, Y_{0:T}^i) = \log \frac{p_{\mathcal{D}}(Y_{0:T}^i)}{L_T^{Y_{0:T}^i}(\theta)} + \mathrm{KL}\left(Q_{\varphi,0:T}^{Y_{0:T}^i} \middle\| \Phi_{\theta,0:T|T}^{Y_{0:T}^i}\right).$$

Define

$$(\widehat{\theta}_{n,T}, \widehat{\varphi}_{n,T}) \in \mathrm{argmin}_{\theta \in \Theta, \varphi \in \Phi} \mathcal{L}_{n,T}(\theta, \varphi). \tag{2}$$

Although this is not the focus of this paper, note that in practice the estimators given by (2) cannot be computed explicitly, in particular when the decoder or the encoder are parameterized by neural networks. In this case, a common approach is to obtain approximate estimators using stochastic gradient descent-based approaches like ADAM Kingma & Ba (2015), see for instance Chagneux et al. (2022) and the references therein. In these settings, the gradient descent is performed using mini batches of trajectories for each parameter update and the unknown expectations are approximated using Monte Carlo methods.

Such a procedure is a so-called $M$-estimation method in the statistical literature. The intuition is that with large data sets, that is when $n$ is large, the ELBO is close to the expected (under the unknown distribution of the data) value of $\varpi$, and the estimated decoding and coding parameters are close to minimizing this expected value. An important body of work in the statistical community has been devoted to develop very general settings in which non asymptotic bounds on the risk of $M$-estimators, referred to as oracle inequalities, can be given, see van de Geer (2000) as early reference, or Wainwright (2019) and the references therein for more recent results. Moreover, oracle inequalities are obviously the only property one can hope for such estimators, the other properties being consequences of the oracle inequality. In the following section, we thus first provide assumptions under which we obtain an oracle inequality for the coding and decoding distributions with parameters defined in (2) and then discuss consequences. Note that though (2) defines the parameters when the algorithm has reached the optimal value, we could relax the definition by minimizing up to some error term that would be added to the upper bound of the oracle inequality.

## 2.3 Examples

We can consider for instance generative models where the transition kernels and emission distributions are Gaussian and parameterized by neural networks.

**Gaussian backward kernels with dense networks.**

- For all $x \in \mathsf{X}$, $x' \mapsto m_\theta(x, x')$ is the Gaussian probability density function with mean $\mu_\theta(x)$, and variance $\Sigma_\theta(x)$ where $(\mu_\theta(x), \Sigma_\theta(x)) = \mathsf{MLP}^\theta(x)$ with $\mathsf{MLP}^\theta$ a dense Multi-layer network with input $x$ and weights given by $\theta$.

- For all $1 \le t \le T$, $x \in \mathsf{X}$, $x' \mapsto q_{\varphi,t-1|t}^{y_{0:T}}(x, x')$ is the Gaussian probability density function with mean $\mu_{\varphi,t-1|t}^{y_{0:T}}(x)$, and variance $\Sigma_{\varphi,t-1|t}^{y_{0:T}}(x)$ where $(\mu_{\varphi,t-1|t}^{y_{0:T}}(x), \Sigma_{\varphi,t-1|t}^{y_{0:T}}(x)) = \mathsf{MLP}_{t-1|t}^{y_{0:T},\varphi}(x)$ with $\mathsf{MLP}_{t-1|t}^{y_{0:T},\varphi}$ a dense Multi-layer network with input $x$ and weights depending on $\varphi$.

Another example where the forward Markov kernels and the backward variational kernels are Gaussian is given by the Chaotic Recurrent Neural Network (CRNN) described in Campbell et al. (2021); Zhao et al. (2022); Chagneux et al. (2022).

**Gaussian backward kernels with recurrent networks.** A natural parameterization is also to use a recurrent neural network which updates an internal state $(s_t)_{t \ge 0}$ from which the backward variational kernels and filtering density are built. For all $t \ge 0$, define $s_t = \mathsf{RNN}^\varphi(s_{t-1}, y_t)$ where $\mathsf{RNN}^\varphi$ is a recurrent neural network, and let $x' \mapsto q_{\varphi,t-1|t}^{y_{0:T}}(x, x')$ be the Gaussian probability density function with mean $\mu_{t-1|t}^{y_{0:T}}$, and variance $\Sigma_{t-1|t}^{y_{0:T}}$ where $(\mu_t, \Sigma_t) = \mathsf{MLP}^\varphi(s_t)$.

## 3 Main results

### 3.1 Assumptions

In this section, we propose a set of assumptions on the kernel densities $m_\theta$ and $q_{\varphi,t|t+1}^{y_{0:T}}$, $0 \le t \le T-1$, and on the conditional densities $g_\theta^y$, under which we are able to prove an oracle inequality. The precise setting of those assumptions is detailed in Appendix B. We discuss in Section 3.3 how they can be applied to specific architectures used in practice. Additional discussions on the assumptions are provided in Appendix G where we prove that usual compact state space models are covered by our theory.

As can be seen in the definition of the ELBO, we shall need to control smoothing expectations both in the coding distribution and in the decoding distribution. In the state space model literature, Assumption H1 is usual for this purpose.

**H1** There exist probability measures $\eta_-$ and $\eta_+$ on $(\mathsf{X}, \mathcal{X})$ and constants $0 < \sigma_- < \sigma_+ < \infty$ such that $\sigma_-\eta_-$ (resp. $\sigma_+\eta_+$) is a uniform lower bound (resp. uniform upper bound) for $\chi$ and $M_\theta$. Similarly, there exist probability measures $\lambda_-$ and $\lambda_+$ on $(\mathsf{X}, \mathcal{X})$ such that for all $y_{0:T} \in \mathsf{Y}^{T+1}$, there exist $\vartheta_-^{y_{0:T}} > 0$ and $\vartheta_+^{y_{0:T}} > 0$ such that $\vartheta_-^{y_{0:T}}\lambda_-$ (resp. $\vartheta_+^{y_{0:T}}\lambda_+$) is a uniform lower bound (resp. uniform upper bound) for $Q_{\varphi,T}^{y_{0:T}}$ and $Q_{\varphi,t|t+1}^{y_{0:T}}$.

In the state space model literature, H2 is usual for the study of asymptotic properties of maximum likelihood estimators.

**H2** For all $y \in \mathsf{Y}$, $\inf_{\theta \in \Theta} \int g_\theta^y(x)\eta_-(\mathrm{d}x) = c_-(y) > 0$ and $\sup_{\theta \in \Theta} \int g_\theta^y(x)\eta_+(\mathrm{d}x) = c_+(y) < \infty$.

More assumptions are needed to manage the complexity of the models and to get a nonasymptotic control of the risk of the estimators. These controls are obtained with Assumptions H3-6.

We constrain the kernels and the conditional densities to be Lipschitz in the parameters with a Lipschitz coefficient depending on the variables.

**H3** There exists $M, G^y, K_{t-1|t}^{y_{0:T}}, K_T^{y_{0:T}}$, such that $m_\theta(x, x')$ is Lipschitz in $\theta$ with Lipschitz coefficient $M(x, x')$, $g_\theta^y(x)$ is Lipschitz in $\theta$ with Lipschitz coefficient $G^y(x)$, $q_{\varphi,t-1|t}^{y_{0:T}}(x, x')$ is Lipschitz in $\varphi$ with Lipschitz coefficient $K_{t-1|t}^{y_{0:T}}(x', x)$, $q_{\varphi,T}^{y_{0:T}}(x)$ is Lipschitz in $\varphi$ with Lipschitz coefficient $K_T^{y_{0:T}}(x)$.

We shall also need Lipschitz properties of the functions comparing logarithms of the backward kernels of the coding and decoding distributions. The precise technical setting of the assumption is given in Appendix B.

**H4** The difference between logarithms of the backward kernels of the coding and of the decoding distributions satisfy Lipschitz properties with respect to the parameters. Moreover, the integral with respect to $\lambda_+$ of this difference is uniformly upper bounded.

We shall need to prove that $\varpi$ is a Lipschitz function of the parameters, and we need an upper bound on the $L^2$-norm of the Lipschitz coefficient. For this purpose, we consider moment assumptions. The precise technical setting of the assumption is given in Appendix B

**H5** There exists $A$ which is an upper bound of several moments involving quantities defined in the previous assumptions.

The last assumption is used to get concentration properties, as usual in the statistical literature to get theoretical guarantees with finite samples. It involves Orlicz norms, the precise definition of which is given in Appendix A.

**H6** There exists $\alpha_*$ and $B > 0$ such that the Orlicz norm of order $\alpha_*$ of several functions is upper bounded with $B$.

### 3.2 Oracle inequalities and consequences

Our main result is an oracle inequality for the risk. The so-called variance term has the usual rate $1/n$ up to $\log n$ terms in the sample size $n$. It is proved to grow as much as $T^3$ in the length $T$ of the sample sequences. We assume that $\Theta$ and $\Phi$ are compact spaces, and that the sum of their diameters is bounded by $d_0$.

**Theorem 3.1.** *Assume that H1-H6 hold. Then, there exist constants $c_0$, $c_1$, $c_2$, $\tilde{D}$ which depend on $\sigma_+$, $\sigma_-$, $\alpha_*$, A, B and $d_0$ only, such that with probability at least $1 - c_0\exp(-c_1\{d_* \log n\}^{1\wedge\alpha_*})$,*

$$\int \varpi(\widehat{\theta}_{n,T}, \widehat{\varphi}_{n,T}, y_{0:T})p_\mathcal{D}(y_{0:T})\mathrm{d}\mu(y_{0:T}) \leq \inf_{\gamma>0} \left\{ (1+\gamma)\mathsf{E}_T + c_2(1+\gamma^{-1})\frac{\tilde{D}d_* T^3}{n}\log(d_* n)(\log n)^{1/\alpha_*} \right\},$$

*where $\mathsf{E}_T = \min_{\theta\in\Theta, \varphi\in\Phi}\int \varpi(\theta, \varphi, y_{0:T})p_\mathcal{D}(y_{0:T})\mathrm{d}\mu(y_{0:T})$ and $d_* = d_\theta + d_\varphi$.*

In Chagneux et al. (2022), under similar assumptions as the strong mixing assumptions of our paper, the authors obtain a linear growth in $T$ as an upper bound on the variational posterior error when the objective is to compute expectations of additive functionals. In our setting we target a more challenging objective i.e. obtaining excess risk bounds and a full control on the divergence between the true data distribution and the estimated one (Corollary 3.2). Indeed, as an intermediate step, the proof of Theorem 3.1 requires similar bounds as the one in Chagneux et al. (2022). The sketch of proof we follow (see below) leads to a growth of order $T^3$. We do not claim that our upper bounds is optimal but it is not surprising that it has a larger growth than the one of Chagneux et al. (2022). Finding another method to prove a bound of better order if possible is an open problem.

In our assumptions, many constants depend on the state dimension and on the dimension of the parameter space. In this paper, we decided to set the focus on the dependency on the number of samples and on $T$. A reason is that tracking the dependency on the dimension is challenging in state space models. Tracking the dependency on $d$ for mixing constants is a contribution on its own for general state spaces without assuming too restrictive conditions on the model. This is the focus of future works.

*Proof.* To prove Theorem 3.1, we use Theorem C.1, which is an improved formulation of (Tang & Yang, 2021, Theorem 3), proved in Appendix C, in which we track the dependency in $n$ and $T$. Theorem C.1 provides an oracle inequality in which the upper bound is a sum of bias term and a variance term. The bias term is $\mathsf{E}_T$. The variance term is the product of a rate in the sample size $n$ and a complexity term. The rate

in $n$ is the usual one, and the complexity term is itself a product of the constants $a_1$ and $D$ involved in the assumptions of Theorem C.1 together with the dimensional constant $d_*$. The main purpose of the remaining of the proof is to understand how $a_1$ and $D$ depend on $T$, the length of the sequences. This requires to provide assumptions that can be verified in specific models such as the models described in Sections 2.3 and 3.3, and to control precisely behaviours of terms involving smoothing (coding and decoding) distributions which is known to be challenging. The detailed proof is given in the Appendix D. We show that $a_1$ may be upper bounded with $CT^2$ for some constant $C > 0$, and that $D$ may be upper bounded with $\tilde{D}T$, resulting in a product of order $T^3$. $\qquad\square$

Note that

$$\int \varpi(\widehat{\theta}_{n,T}, \widehat{\varphi}_{n,T}, y_{0:T}) p_{\mathcal{D}}(y_{0:T}) \mathrm{d}\mu(y_{0:T}) = \mathrm{KL}\left(P_{\mathcal{D}} \middle\| P_{\widehat{\theta}_{n,T}}^Y\right) + \mathbb{E}_{P_{\mathcal{D}}} \mathrm{KL}\left(Q_{\widehat{\varphi}_{n,T},0:T}^{Y_{0:T}^1} \middle\| \Phi_{\widehat{\theta}_{n,T},0:T|T}^{Y_{0:T}^1}\right).$$

If the upper bound in Theorem 3.1 is small, then the distribution $P_{\mathcal{D}}$ of the observations is well approximated by the decoding observational distribution $P_{\widehat{\theta}_{n,T}}^Y$, and the decoding distribution of the latent state distribution given data $\Phi_{\widehat{\theta}_{n,T},0:T|T}^{Y_{0:T}^1}$ is also in average well approximated by the coding distribution $Q_{\widehat{\varphi}_{n,T},0:T}^{Y_{0:T}^1}$.

In the same way,

$$\mathsf{E}_T = \min_{\theta \in \Theta, \varphi \in \Phi} \left\{ \mathrm{KL}\left(P_{\mathcal{D}} \middle\| P_\theta^Y\right) + \mathbb{E}_{P_{\mathcal{D}}} \mathrm{KL}\left(Q_{\varphi,0:T}^{Y_{0:T}^1} \middle\| \Phi_{\theta,0:T|T}^{Y_{0:T}^1}\right)\right\}.$$

In case the data follows a state space distribution given by some decoding distribution, that is if there exists $\theta^* \in \Theta$ such that $P_{\mathcal{D}} = P_{\theta^*}^Y$, the oracle inequality in Theorem 3.1 becomes, by taking $\theta = \theta^*$ to upper bound $\mathsf{E}_T$,

$$\mathrm{KL}\left(P_{\theta^*}^Y \middle\| P_{\widehat{\theta}_{n,T}}^Y\right) + \mathbb{E}_{P_{\theta^*}^Y} \mathrm{KL}\left(Q_{\widehat{\varphi}_{n,T},0:T}^{Y_{0:T}^1} \middle\| \Phi_{\widehat{\theta}_{n,T},0:T|T}^{Y_{0:T}^1}\right) \leq (1+\gamma) \min_{\varphi \in \Phi} \mathbb{E}_{P_{\theta^*}^Y} \mathrm{KL}\left(Q_{\varphi,0:T}^{Y_{0:T}^1} \middle\| \Phi_{\theta^*,0:T|T}^{Y_{0:T}^1}\right)$$
$$+ c_2(1+\gamma^{-1}) \frac{Dd_* T^3}{n} \log(d_* n)(\log n)^{1/\alpha_*} \quad (3)$$

for any $\gamma > 0$. In the following corollary, we assume that the coding backward kernels are chosen such that they are good approximations of the backward decoding kernels in Kullback-Leibler divergence.

**H7** There exists $\epsilon > 0$, such that for all $\theta \in \Theta$ there exists $\varphi \in \Phi$ such that for all $y_{0:T} \in \mathsf{Y}^{T+1}$,

$$\mathrm{KL}\left(Q_{\varphi,T}^{y_{0:T}} \middle\| \Phi_{\theta^*,T}^{y_{0:T}}\right) \leq \epsilon$$

and for all $1 \leq t \leq T$,

$$\mathrm{KL}\left(Q_{\varphi,t-1|t}^{y_{0:T}} \middle\| B_{\theta,t-1|t}^{y_{0:t-1}}\right) \leq \epsilon.$$

**Corollary 1.** *Assume there exists $\theta^* \in \Theta$ such that $P_{\mathcal{D}} = P_{\theta^*}^Y$. Assume moreover H7. Then under the same assumptions as in Theorem 3.1, for the constants $c_0$, $c_1$, $c_2$, $D$ in Theorem 3.1, with probability at least $1 - c_0 \exp(-c_1\{d_* \log n\}^{1 \wedge \alpha_*})$, for any $\gamma > 0$,*

$$\mathrm{KL}\left(P_{\theta^*}^Y \middle\| P_{\widehat{\theta}_{n,T}}^Y\right) + \mathbb{E}_{P_{\theta^*}^Y} \mathrm{KL}\left(Q_{\widehat{\varphi}_{n,T},0:T}^{Y_{0:T}^1} \middle\| \Phi_{\widehat{\theta}_{n,T},0:T|T}^{Y_{0:T}^1}\right) \leq (1+\gamma)(T+1)\epsilon + c_2(1+\gamma^{-1}) \frac{Dd_* T^3}{n} \log(d_* n)(\log n)^{1/\alpha_*}.$$

When the data distribution is given by a state space model, Corollary 1 provides an upper bound for the Kullback-Leibler divergence between the data distribution and its estimator and between the variational posterior and the estimated state space posterior distributions. This result sheds additional light on the quality of variational reconstruction in state space models with respect to (Chagneux et al., 2022, Proposition 3). In (Chagneux et al., 2022, Proposition 3), the authors provided upper bounds on the error between conditional expectations of state functionals under the true posterior distribution and under its variational approximation. In both settings, designing coding backward kernels that are good approximations of the true backward decoding kernels is enough to obtain quantitative controls on the reconstruction error.

*Proof.* The result follows from equation 3, H7 and the fact that for any $\theta \in \Theta$ and $\varphi \in \Phi$, for any $y_{0:T}$,

$$\mathrm{KL}\left(Q_{\varphi,0:T}^{y_{0:T}}\Big\|\Phi_{\theta^*,0:T|T}^{y_{0:T}}\right) = \sum_{t=1}^{T}\mathrm{KL}\left(Q_{\varphi,t-1|t}^{y_{0:T}}\Big\|B_{\theta,t-1|t}^{y_{0:t-1}}\right) + \mathrm{KL}\left(Q_{\varphi,T}^{y_{0:T}}\Big\|\Phi_{\theta^*,T}^{y_{0:T}}\right).$$

$\square$

### 3.3  Applications

In this section, we consider generative models where the transition kernels and emission distributions are Gaussian in various classical settings. We show that under weak assumptions on these models, some assumptions of our main results hold. Establishing that all assumptions are satisfied in general settings, i.e. without very specific assumptions on the architectures, is a more challenging problem.

We prove in Appendix G that H1 holds in particular for compact state spaces. We also prove that the functions $h_{t,\theta,\varphi}^{y_{0:T}}$ are upper-bounded explicitly, and that $\phi_{\theta,t}^{y_{0:t}}$ and $b_{\theta,t-1|t}^{y_{0:t-1}}$ are lower and upper-bounded explicitly. This allows to obtain explicit constants in H4. Providing additional comments on the assumptions requires assumptions on the observation space or on the dependency of the variational distributions on the observations. When the observation space is compact we can also obtain a uniform control with respect to the observations of these upper bounds which is crucial to check H5 and H6.

**Gaussian backward kernels with dense networks.**     We consider a generative model where the transition kernels and emission distributions are Gaussian and parameterized by dense networks following Section 2.3.

- For all $x \in \mathsf{X}$, $x' \mapsto m_\theta(x, x')$ is the Gaussian probability density function with mean $\mu_\theta(x)$, and variance $\Sigma_\theta(x)$ where $(\mu_\theta(x), \Sigma_\theta(x)) = \mathsf{MLP}^\theta(x)$ with $\mathsf{MLP}^\theta$ a dense Multi-layer network with input $x$ and weights given by $\theta$. In this case, if the output layer of $\mathsf{MLP}^\theta$ is such that $\mu_\theta$ is bounded and $\underline{\Sigma} \leq \Sigma_\theta^{-1}(x) \leq \overline{\Sigma}$ (i.e. $\Sigma_\theta^{-1}(x) - \underline{\Sigma}$ and $\overline{\Sigma} - \Sigma_\theta^{-1}(x)$ are positive semi-definite matrices) for all $x \in \mathsf{X}$, then there exist constants $\underline{c}, \overline{c}$ such that for all $x, x' \in \mathsf{X}$,

$$\underline{c}\exp\left(-\overline{\lambda}x^\top x\right) \leq m_\theta(x', x) \leq \overline{c}\exp\left(-\underline{\lambda}\alpha(x)\right) ,$$

where $\underline{\lambda}$ is the smallest eigenvalue of $\underline{\Sigma}$ and $\overline{\lambda}$ is the largest eigenvalue of $\overline{\Sigma}$ and where

$$\alpha(x) = \frac{1}{2}\left((\|x\| - M)^2\mathbb{1}_{\|x\|\geq M} + (\|x\| - m)^2\mathbb{1}_{\|x\|\leq m} + (M - m)^2\mathbb{1}_{m\leq\|x\|\leq M}\right) ,$$

with $m = \inf_{x\in\mathsf{X},\theta\in\Theta}\|\mu_\theta(x)\|$ and $M = \sup_{x\in\mathsf{X},\theta\in\Theta}\|\mu_\theta(x)\|$. This implies that H1 holds. In order to check H3, if we assume also that for all $x \in \mathsf{X}$, $\theta \mapsto \mu_\theta(x)$ and $\theta \mapsto \Sigma_\theta^{-1}(x)$ are continuously differentiable and that $\Theta$ is compact then there exists $M$ such that for all $\theta, \theta' \in \Theta$ and $x, x' \in \mathsf{X}$,

$$|m_\theta(x, x') - m_{\theta'}(x, x')| \leq M(x, x')\|\theta - \theta'\|_2 .$$

We can check H4 for $\log b_{\theta,t-1|t}^{y_{0:t-1}}$, as other items can be verified following the same steps. Assuming that $b_{\theta,t-1|t}^{y_{0:t-1}}(x, \cdot)$ is a Gaussian probability density with mean $\mu_{\theta,t-1|t}^{y_{0:t-1}}(x)$ and variance $\Sigma_{\theta,t-1|t}^{y_{0:t-1}}(x)$. Under similar regularity assumptions on the networks providing $\mu_{\theta,t-1|t}^{y_{0:t-1}}(x)$ and $\Sigma_{\theta,t-1|t}^{y_{0:t-1}}(x)$, when $\Theta$ is compact, H4 holds.

- For all $1 \leq t \leq T$, $x \in \mathsf{X}$, $x' \mapsto q_{\varphi,t-1|t}^{y_{0:T}}(x, x')$ is the Gaussian probability density function with mean $\mu_{\varphi,t-1|t}^{y_{0:T}}(x)$, and variance $\Sigma_{\varphi,t-1|t}^{y_{0:T}}(x)$ where $(\mu_{\varphi,t-1|t}^{y_{0:T}}(x), \Sigma_{\varphi,t-1|t}^{y_{0:T}}(x)) = \mathsf{MLP}_{t-1|t}^{y_{0:T},\varphi}(x)$ with $\mathsf{MLP}_{t-1|t}^{y_{0:T},\varphi}$ a dense Multi-layer network with input $x$ and weights depending on $\varphi$. In this case, is the output layer of $\mathsf{MLP}_{t-1|t}^{y_{0:T},\varphi}$ is such that $\mu_{\varphi,t-1|t}^{y_{0:T}}$ is bounded and $\underline{\Sigma}_{t-1|t}^{y_{0:T}} \leq (\Sigma_{\varphi,t-1|t}^{y_{0:T}}(x))^{-1} \leq \overline{\Sigma}_{t-1|t}^{y_{0:T}}$

(i.e. $(\Sigma_{\varphi,t-1|t}^{y_{0:T}}(x))^{-1} - \underline{\Sigma}_{t-1|t}^{y_{0:T}}$ and $\overline{\Sigma}_{t-1|t}^{y_{0:T}} - (\Sigma_{\varphi,t-1|t}^{y_{0:T}}(x))^{-1}$ are positive semi-definite matrices) for all $x \in \mathsf{X}$, then there exist constants $\underline{c}_{t-1|t}^{y_{0:T}}$, $\overline{c}_{t-1|t}^{y_{0:T}}$ such that for all $x, x' \in \mathsf{X}$,

$$\underline{c}_{t-1|t}^{y_{0:T}} \exp\left(-\overline{\lambda}_{t-1|t}^{y_{0:T}} x^\top x\right) \leq q_{\varphi,t-1|t}^{y_{0:T}}(x', x) \leq \overline{c}_{t-1|t}^{y_{0:T}} \exp\left(-\underline{\lambda}_{t-1|t}^{y_{0:T}} \beta(x)\right) ,$$

where $\underline{\lambda}_{t-1|t}^{y_{0:T}}$ is the smallest eigenvalue of $\underline{\Sigma}_{t-1|t}^{y_{0:T}}$ and $\overline{\lambda}_{t-1|t}^{y_{0:T}}$ is the largest eigenvalue of $\overline{\Sigma}_{t-1|t}^{y_{0:T}}$ and where

$$\beta(x) = \frac{1}{2}\left((\|x\| - M_{t-1|t}^{y_{0:T}})^2 \mathbb{1}_{\|x\| \geq M_{t-1|t}^{y_{0:T}}} + (\|x\| - m_{t-1|t}^{y_{0:T}})^2 \mathbb{1}_{\|x\| \leq m_{t-1|t}^{y_{0:T}}}\right.$$
$$\left. + (M_{t-1|t}^{y_{0:T}} - m_{t-1|t}^{y_{0:T}})^2 \mathbb{1}_{m_{t-1|t}^{y_{0:T}} \leq \|x\| \leq M_{t-1|t}^{y_{0:T}}}\right) ,$$

with $m_{t-1|t}^{y_{0:T}} = \inf_{x \in \mathsf{X}} \|\mu_{t-1|t}^{y_{0:T}}(x)\|$ and $M_{t-1|t}^{y_{0:T}} = \sup_{x \in \mathsf{X}} \|\mu_{t-1|t}^{y_{0:T}}(x)\|$. Similar assumptions can be used for $q_{\varphi,T}^{y_{0:T}}$ using dense neural networks with bounded output. Under similar regularity assumptions on $\mu_{\varphi,t-1|t}^{y_{0:T}}$, and $\Sigma_{\varphi,t-1|t}^{y_{0:T}}$ than for $\mu_\theta$, and variance $\Sigma_\theta$, we may prove that H3 holds when $\Phi$ is compact.

**Gaussian backward kernels with recurrent networks.** A natural parameterization is also to use a recurrent neural network which updates an internal state $(s_t)_{t \geq 0}$ from which the backward variational kernels and filtering density are built. For all $t \geq 0$, define $s_t = \mathsf{RNN}^\varphi(s_{t-1}, y_t)$ where $\mathsf{RNN}^\varphi$ is a recurrent neural network, and let $x' \mapsto q_{\varphi,t-1|t}^{y_{0:T}}(x, x')$ be the Gaussian probability density function with mean $\mu_{t-1|t}^{y_{0:T}}$, and variance $\Sigma_{t-1|t}^{y_{0:T}}$ where $(\mu_t, \Sigma_t) = \mathsf{MLP}^\varphi(s_t)$. If the network $\mathsf{MLP}^\varphi$ is bounded similarly as in the dense neural network case, then the backward variational kernels satisfy H1.

**Functional autoregressive models.** The discussion on neural networks also indicates that the assumptions can be verified for some classical statistical models. Assume for instance that $\mathsf{X} = \mathbb{R}$ and that for all $\theta \in \Theta$, $x \in \mathsf{X}$, $x' \mapsto m_\theta(x, x')$ is the Gaussian probability density function with mean $f_\theta(x)$, and variance $\sigma_\theta^2(x)$. Then, H1 holds for $m_\theta$ when $-\infty < \inf_{x \in \mathsf{X}, \theta \in \Theta} f_\theta(x) \leq \sup_{x \in \mathsf{X}, \theta \in \Theta} f_\theta(x) < \infty$ and $-\infty < \inf_{x \in \mathsf{X}, \theta \in \Theta} \sigma_\theta(x) \leq \sup_{x \in \mathsf{X}, \theta \in \Theta} \sigma_\theta(x) < \infty$.

**Gaussian emission densities.** Assume that at each time $t \geq 0$, $Y_t = h_\theta(X_t) + \varepsilon_t$, where $\{\varepsilon_t\}_{t \geq 0}$ are independent Gaussian random variables. Assume also that $h_\theta(X_t) = \mathsf{MLP}^\theta(X_t)$ where $\mathsf{MLP}^\theta$ is a dense neural network with bounded output layer, then H2 holds. Assume that for all $x \in \mathsf{X}$, $\theta \mapsto h_\theta(x)$ is continuously differentiable and that $\Theta$ is compact, for all $y \in \mathsf{Y}$, there exists $G^y$ such that for all $\theta, \theta' \in \Theta$ and $x \in \mathsf{X}$,

$$|g_\theta^y(x) - g_{\theta'}^y(x)| \leq G^y(x)\|\theta - \theta'\|_2 ,$$

which means that H3 holds for the emission distributions.

## 4 Discussion

In this paper, we used a backward decomposition of variational posterior distributions to propose the first theoretical results for variational autoencoders (VAE) applied to general state space models. Under strong mixing assumptions on the state space model and on the variational distribution, we provide in particular an oracle inequality and an upper bound for the Kullback-Leibler divergence between the data distribution and its estimator.

Although these results are the first theoretical guarantees for VAE in the context of state space models, we believe that this is the first step to solve challenging open problems. First, in order to cover a wider variety of applications, weakening the strong mixing assumptions, for instance using local Doeblin assumptions, would be very interesting although it is a challenge when analyzing the stability of smoothing distributions. Another research direction is to understand how our results can be extended in settings where the observations are processed online, i.e. in cases where the parameters are updated when new observations are received but never stored. To the best of our knowledge, online variational estimation has recently been explored with new methodologies but without any theoretical guarantees.

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

# A    Notations.

In the following, for all measures $\lambda$ and $\eta$ on $(\mathsf{X}, \mathcal{X})$ and all transition kernels $K$ we consider the following notations. For all measurable sets $A \subset \mathsf{X} \times \mathsf{X}$, $\lambda \otimes \eta(A) = \int \mathbb{1}_A(x, x')\lambda(\mathrm{d}x)\eta(\mathrm{d}x')$ and $\lambda \otimes K(A) = \int \mathbb{1}_A(x, x')\lambda(\mathrm{d}x)K(x, \mathrm{d}x')$, for all measurable sets $B \subset \mathsf{X}$, $\lambda K(B) = \int \lambda(\mathrm{d}x)\mathbb{1}_B(x')K(x, \mathrm{d}x')$, and for all real-valued measurable functions $h$ on $(\mathsf{X}, \mathcal{X})$, $\lambda(h) = \int \lambda(\mathrm{d}x)h(x)$. For all measurable functions $h_1, h_2$, we write $h_1 \otimes h_2 : (x, x') \mapsto h_1(x)h_2(x')$.

For all $\alpha > 0$, define on $\mathbb{R}_+$ the function $\psi_\alpha : x \mapsto \exp(x^\alpha) - 1$.

For all real-valued random variables $X$, define the Orlicz norm of order $\alpha$ by

$$\|X\|_{\psi_\alpha} = \inf_{\lambda > 0} \{\mathbb{E}\left[\psi_\alpha(|X|/\lambda)\right] \leq 1\} .$$

For all probability measures $P$ and $Q$ defined on the same probability space, $\|P - Q\|_{\mathrm{tv}}$ will denote the total variation norm between $P$ and $Q$.

# B    Assumptions

In this section, we give the precise setting of the assumptions.

**H1** There exist probability measures $\eta_-$ and $\eta_+$ on $(\mathsf{X}, \mathcal{X})$ and constants $0 < \sigma_- < \sigma_+ < \infty$ such that for all $\theta \in \Theta$, $x \in \mathsf{X}$, all measurable set $A$,

$$\sigma_-\eta_-(A) \leq \chi(A) \leq \sigma_+\eta_+(A)$$

and

$$\sigma_-\eta_-(A) \leq M_\theta(x, A) \leq \sigma_+\eta_+(A) .$$

There exist probability measures $\lambda_-$ and $\lambda_+$ on $(\mathsf{X}, \mathcal{X})$ such that for all $y_{0:T} \in \mathsf{Y}^{T+1}$, there exist $\vartheta_-^{y_{0:T}} > 0$ and $\vartheta_+^{y_{0:T}} > 0$ such that for all $\varphi \in \Phi$, $t \geq 0$, $x \in \mathsf{X}$, all measurable set $A$,

$$\vartheta_-^{y_{0:T}}\lambda_-(A) \leq Q_{\varphi, t|t+1}^{y_{0:T}}(x, A) \leq \vartheta_+^{y_{0:T}}\lambda_+(A) .$$

In addition, for all $\varphi \in \Phi$, all $y_{0:T} \in \mathsf{Y}^{T+1}$, and all measurable set $A$,

$$\vartheta_-^{y_{0:T}}\lambda_-(A) \leq Q_{\varphi, T}^{y_{0:T}}(A) \leq \vartheta_+^{y_{0:T}}\lambda_+(A).$$

**H2** For all $y \in \mathsf{Y}$, $\inf_{\theta \in \Theta} \int g_\theta^y(x)\eta_-(\mathrm{d}x) = c_-(y) > 0$ and $\sup_{\theta \in \Theta} \int g_\theta^y(x)\eta_+(\mathrm{d}x) = c_+(y) < \infty$.

We consider also the following notation $\sup_{\theta \in \Theta} g_\theta^{y_t} = \bar{g}^{y_t}$ and $\inf_{\theta \in \Theta} g_\theta^{y_t} = \underline{g}^{y_t}$.

**H3** There exists $M$ such that for all $\theta, \theta' \in \Theta$ and $x, x' \in \mathsf{X}$,

$$|m_\theta(x, x') - m_{\theta'}(x, x')| \leq M(x, x')\|\theta - \theta'\|_2 \ .$$

For all $1 \leq t \leq T$, $y_{0:T}$, there exists $K_{t-1|t}^{y_{0:T}}$ such that for all $\varphi, \varphi' \in \Phi$ and $x, x' \in \mathsf{X}$,

$$\left|q_{\varphi,t-1|t}^{y_{0:T}}(x, x') - q_{\varphi',t-1|t}^{y_{0:T}}(x, x')\right| \leq K_{t-1|t}^{y_{0:T}}(x', x)\|\varphi - \varphi'\|_2 \ .$$

In addition, there exists $K_T^{y_{0:T}}$ such that for all $\varphi, \varphi' \in \Phi$ and $x \in \mathsf{X}$,

$$\left|q_{\varphi,T}^{y_{0:T}}(x) - q_{\varphi',T}^{y_{0:T}}(x)\right| \leq K_T^{y_{0:T}}(x)\|\varphi - \varphi'\|_2 \ .$$

For all $y \in \mathsf{Y}$, there exists $G^y$ such that for all $\theta, \theta' \in \Theta$ and $x \in \mathsf{X}$,

$$|g_\theta^y(x) - g_{\theta'}^y(x)| \leq G^y(x)\|\theta - \theta'\|_2 \ .$$

Define, for $1 \leq t \leq T - 1$,

$$h_{t,\theta,\varphi}^{y_{0:T}}(x_{t-1}, x_t) = \log q_{\varphi,t-1|t}^{y_{0:T}}(x_t, x_{t-1}) - \log b_{\theta,t-1|t}^{y_{0:t-1}}(x_t, x_{t-1}) \tag{4}$$

and, by convention, $h_{T,\theta,\varphi}^{y_{0:T}}(x_{T-1}, x_T) = \log q_{\varphi,T-1|T}^{y_{0:T}}(x_T, x_{T-1}) - \log b_{\theta,T-1|T}^{y_{0:T-1}}(x_T, x_{T-1}) + \log q_{\varphi,T}^{y_{0:T}}(x_T) - \log \phi_{\theta,T}^{y_{0:T}}(x_T)$.

**H4** For all $y_{0:T} \in \mathsf{Y}^{T+1}$ and all $0 \leq t \leq T$,

$$\sup_{\theta \in \Theta, \varphi \in \Phi} \left\| \int \lambda_+(\mathrm{d}x) \left| h_{t,\theta,\varphi}^{y_{0:T}}(x, \cdot) \right| \right\|_\infty = v_t^{y_{0:T}} < \infty \ ,$$

and for all $\theta, \theta' \in \Theta$, $\varphi, \varphi' \in \Phi$, $1 \leq t \leq T$,

$$\int \lambda_+ \otimes \lambda_+(\mathrm{d}x\mathrm{d}x') \left| \log q_{\varphi,t-1|t}^{y_{0:T}}(x, x') - \log q_{\varphi',t-1|t}^{y_{0:T}}(x, x') \right| \leq c_{1,t}^{y_{0:T}} \|\varphi - \varphi'\|_2 \ ,$$

$$\int \lambda_+ \otimes \lambda_+(\mathrm{d}x\mathrm{d}x') \left| \log b_{\theta,t-1|t}^{y_{0:t-1}}(x, x') - \log b_{\theta',t-1|t}^{y_{0:t-1}}(x, x') \right| \leq c_{2,t}^{y_{0:t-1}} \|\theta - \theta'\|_2 \ ,$$

$$\int \lambda_+(\mathrm{d}x) \left| \log q_{\varphi,T}^{y_{0:T}}(x) - \log q_{\varphi',T}^{y_{0:T}}(x) \right| \leq c_{3,T}^{y_{0:T}} \|\varphi - \varphi'\|_2 \ ,$$

$$\int \lambda_+(\mathrm{d}x) \left| \log \phi_{\theta,T}^{y_{0:T}}(x) - \log \phi_{\theta',T}^{y_{0:T}}(x) \right| \leq c_{4,T}^{y_{0:T}} \|\theta - \theta'\|_2 \ ,$$

where $\lambda_+$ is defined in H1.

**H5** There exists $A$ such that the following inequalities are satisfied.

$$\mathbb{E}\left[\left(\vartheta_+^{Y_{0:T}} c_{3,T}^{Y_{0:T}}\right)^2\right] \leq A \ , \ \mathbb{E}\left[\left(\vartheta_+^{Y_{0:T}} c_{4,T}^{Y_{0:T}}\right)^2\right] \leq A \ ,$$

for all $0 \leq t \leq T$,

$$\mathbb{E}\left[\frac{\mu(G^{Y_t})^2}{c_-(Y_t)^2}\right] \leq A \ , \ \mathbb{E}\left[\left((\vartheta_+^{Y_{0:T}})^2 c_{1,t}^{Y_{0:T}}\right)^2\right] \leq A \ ,$$

for all $1 \leq t \leq T$,

$$\mathbb{E}\left[\left((\vartheta_+^{Y_{0:T}})^2 c_{2,t}^{Y_{0:t-1}}\right)^2\right] \leq A \ , \ \mathbb{E}\left[\frac{\eta_+ \otimes \mu(M \otimes \bar{g}^{Y_{t-1}} \bar{g}^{Y_t})^2}{c_-(Y_{t-1})^2 c_-(Y_t)^2}\right] \leq A \ ,$$

$$\mathbb{E}\left[\left(\vartheta_+^{Y_{0:T}} \sum_{s=t-1}^{T} \lambda_+ \otimes \lambda_+(K_{s|s+1}^{y_{0:T}}) \rho(Y_{0:T})^{s-t}\right)^2\right] \leq A\,,$$

where for all $y_{0:T}$, $\rho(y_{0:T}) = 1 - \vartheta_-^{y_{0:T}}$, for all $0 \leq s, t \leq T$,

$$\mathbb{E}\left[\frac{c_+(Y_t)^2 \mu(G^{Y_s})^2}{c_-(Y_t)^2 c_-(Y_s)^2}\right] \leq A\,,$$

and for all $0 \leq t \leq T$, all $1 \leq s \leq T$,

$$\mathbb{E}\left[\left(\frac{c_+(Y_t)\eta_+ \otimes \mu(M \otimes \bar{g}^{Y_{s-1}} \bar{g}^{Y_s})}{c_-(Y_{s-1})c_-(Y_s)c_-(Y_t)}\right)^2\right] \leq A\,.$$

**H6** There exists $\alpha_*$ and $B > 0$ such that for all $T \geq 1$,

$$\|\log p_{\mathcal{D}}(Y_{0:T})\|_{\psi_{\alpha_*}} \leq BT \quad \text{and} \quad \left\|(\vartheta_+^{Y_{0:T}})^2 \cdot \sup_{\theta,\varphi,\chi} \sum_{t=1}^{T} \lambda_+ \otimes \lambda_+\left(\left|h_{t,\theta,\varphi}^{y_{0:T}}\right|\right)\right\|_{\psi_{\alpha_*}} \leq BT\,,$$

and for all $0 \leq t \leq T$,

$$\||\log c_+(Y_t)| \vee |\log c_-(Y_t)|\|_{\psi_{\alpha_*}} \leq B\,.$$

## C  An oracle inequality adapted from Tang & Yang (2021)

We propose an alternative formulation of Theorem 3 in Tang & Yang (2021) in which we provide the precise behavior of the constant in the variance term. To avoid introducing too many new notations, we formulate the results of Tang & Yang (2021) choosing the observation to be $Y_{0:T}$, the latent variables to be $X_{0:T}$ in our setting.

**Condition A.**  There exist $a_1 > 0$ and a function $b$ such that for all $\theta \in \Theta$, $\theta' \in \Theta$, $\varphi \in \Phi$, $\varphi' \in \Phi$, $y_{0:T} \in \mathsf{Y}^{T+1}$,

$$|\varpi(\theta,\varphi,y_{0:T}) - \varpi(\theta',\varphi',y_{0:T})| \leq b(y_{0:T})\|(\theta,\varphi) - (\theta',\varphi')\|_2\,,$$

with $\mathbb{E}[b^2(Y_{0:T})] \leq a_1$.

**Assumption A.**  There exist $\alpha_* > 0$ and $D > 0$ such that

$$\left\|\sup_{\theta,\varphi}\left\{\left|\log \frac{L_T^{Y_{0:T}}(\theta)}{p_{\mathcal{D}}(Y_{0:T})}\right| + \mathrm{KL}\left(Q_{\varphi,0:T}^{Y_{0:T}} \middle\| \phi_{\theta,T}^{Y_{0:T}}\right)\right\}\right\|_{\psi_{\alpha_*}} \leq D\,. \tag{5}$$

**Theorem C.1.** *Assume that $\Theta$ and $\Phi$ are compact spaces and that the sum of their diameter is upper bounded by $d_0$. Assume moreover that Condition A and Assumption A hold. Then, there exist constants $c_0$, $c_1$, which depend on $d_0$, $a_1$ and $\alpha_*$, and a universal constant $c_2$, such that with probability at least $1 - c_0\exp(-c_1\{d_* \log n\}^{1 \wedge \alpha_*})$,*

$$\int \varpi(\widehat{\theta}_{n,T}, \widehat{\varphi}_{n,T}, y_{0:T}) p_{\mathcal{D}}(y_{0:T}) \mathrm{d}\mu(y_{0:T}) \leq \inf_{\gamma > 0}\left\{(1+\gamma)\mathsf{E}_T + c_2(1+\gamma^{-1})\frac{a_1 D d_*}{n}\log(d_* n)(\log n)^{1/\alpha_*}\right\}\,,$$

*where $\mathsf{E}_T = \min_{\theta \in \Theta, \varphi \in \Phi} \int \varpi(\theta,\varphi,y_{0:T}) p_{\mathcal{D}}(y_{0:T}) \mathrm{d}\mu(y_{0:T})$ and $d_* = d_\theta + d_\varphi$.*

*Proof.* We follow the proof of (Tang & Yang, 2021, Theorem 3), in which we track the dependencies of the constants with respect to $a_1$. In (Tang & Yang, 2021, Lemma 14), a multiplicative term $\sqrt{a_1}$ is required on the r.h.s. of the inequality. Then on page 24 third line the inequality needs again $\sqrt{a_1}$ on the r.h.s., and the end of the proof follows by multiplying $\delta_n$ by $\sqrt{a_1}$. We obtain that in (Tang & Yang, 2021, Theorem 3), their constant $c_2$ is proportional to $a_1$. $\qquad\square$

## D  Proof of Theorem 3.1

First, Assumption A of Theorem 3.1 holds with $D = \tilde{D}T$ for some positive constant $\tilde{D}$ depending on $B$. This is a consequence of the first point in H6, Proposition E.4 and Proposition E.5.

We now prove that Condition A of Theorem 3.1 holds i.e. that for all $\theta$, $\theta'$, $\varphi$, $\varphi'$, $y_{0:T}$,

$$\Delta(\theta, \theta', \varphi, \varphi', y_{0:T}) = |\varpi(\theta, \varphi, y_{0:T}) - \varpi(\theta', \varphi', y_{0:T})| \leq b(y_{0:T}) \|(\theta, \varphi) - (\theta', \varphi')\|_2,$$

with $a_1 \leq CT^2$ for some $C > 0$. Write, for all $\theta$, $\varphi_1$, $\varphi_2$, $y_{0:T}$,

$$\mathcal{E}^{y_{0:T}}(\theta, \varphi_1, \varphi_2) = \mathbb{E}_{q^{y_{0:T}}_{\varphi_1,0:T}} \left[ \log \frac{q^{y_{0:T}}_{\varphi_2,0:T}(X_{0:T})}{\phi^{y_{0:T}}_{\theta,0:T|T}(X_{0:T})} \right] .$$

Note that

$$\Delta(\theta, \theta', \varphi, \varphi', y_{0:T}) \leq |\ell^{y_{0:T}}_T(\theta) - \ell^{y_{0:T}}_T(\theta')| + |\mathcal{E}^{y_{0:T}}(\theta, \varphi, \varphi) - \mathcal{E}^{y_{0:T}}(\theta', \varphi', \varphi')| .$$

Write

$$|\mathcal{E}^{y_{0:T}}(\theta, \varphi) - \mathcal{E}^{y_{0:T}}(\theta', \varphi')| \leq \Delta_1(\theta, \varphi, \varphi', y_{0:T}) + \Delta_2(\theta, \theta', \varphi, \varphi', y_{0:T}),$$

where

$$\Delta_1(\theta, \varphi, \varphi', y_{0:T}) = |\mathcal{E}^{y_{0:T}}(\theta, \varphi, \varphi) - \mathcal{E}^{y_{0:T}}(\theta, \varphi', \varphi)| ,$$
$$\Delta_2(\theta, \theta', \varphi, \varphi', y_{0:T}) = |\mathcal{E}^{y_{0:T}}(\theta, \varphi', \varphi) - \mathcal{E}^{y_{0:T}}(\theta', \varphi', \varphi')| .$$

Therefore,

$$\Delta(\theta, \theta', \varphi, \varphi', y_{0:T}) \leq |\ell^{y_{0:T}}_T(\theta) - \ell^{y_{0:T}}_T(\theta')| + \Delta_1(\theta, \varphi, \varphi', y_{0:T}) + \Delta_2(\theta, \theta', \varphi, \varphi', y_{0:T}) .$$

By Proposition E.1, Proposition E.2 and Proposition E.3, we get that for all $\theta$, $\theta'$, $\varphi$, $\varphi'$, and all $y_{0:T}$,

$$\Delta(\theta, \theta', \varphi, \varphi', y_{0:T}) \leq (\kappa_1(y_{0:T}) + \kappa_4(y_{0:T})) \|\theta - \theta'\|_2 + (\kappa_2(y_{0:T}) + \kappa_3(y_{0:T})) \|\varphi - \varphi'\|_2,$$

where

$$\kappa_1(y_{0:T}) = \frac{\sigma_+ \eta_+(G^{y_0})}{\sigma_- c_-(y_0)} + \sum_{t=1}^T \frac{\sigma_+}{\sigma_- c_-(y_t)} \left\{ c_+(y_t) L_{t-1}(y_{0:t-1}) + \frac{\eta_+ \otimes \mu(M \cdot \bar{g}^{y_{t-1}} \bar{\otimes} g^{y_t})}{\sigma_- c_-(y_{t-1})} + \eta_+(G^{y_t}) \right\} , \quad (6)$$

with $M \cdot \bar{g}^{y_{t-1}} \otimes \bar{g}^{y_t}(x, x') = M(x, x') \bar{g}^{y_{t-1}}(x) \bar{g}^{y_t}(x')$, and for all $t$,

$$L_t(y_{0:t}) = \frac{4\sigma_+^2}{\sigma_-^2} \sum_{s=0}^t \varepsilon^{t-s} \frac{1}{c_-(y_s)} \left\{ \frac{1}{\sigma_- c_-(y_{s-1})} \eta_+ \otimes \mu \left( M \cdot \bar{g}^{y_{s-1}} \otimes \bar{g}^{y_s} \right) + \mu(G^{y_s}) \right\} , \quad (7)$$

with $\varepsilon = 1 - \sigma_-/\sigma_+$,

$$\kappa_2(y_{0:T}) = (\vartheta_+^{y_{0:T}})^3 \sum_{t=1}^T \upsilon_t^{y_{0:T}} \sum_{s=t-1}^T \lambda_+ \otimes \lambda_+(K^{y_{0:T}}_{s|s+1}) \rho(y_{0:T})^{s-t} , \quad (8)$$

$$\kappa_3(y_{0:T}) = \vartheta_+^{y_{0:T}} \left( \vartheta_+^{y_{0:T}} \sum_{t=1}^T c^{y_{0:T}}_{1,t} + c^{y_{0:T}}_{3,T} \right) , \quad (9)$$

and

$$\kappa_4(y_{0:T}) = \vartheta_+^{y_{0:T}} \left( \vartheta_+^{y_{0:T}} \sum_{t=1}^T c^{y_{0:t-1}}_{2,t} + c^{y_{0:T}}_{4,t} \right) , \quad (10)$$

in which $\upsilon_t^{y_{0:T}}$, $c^{y_{0:T}}_{1,t}$, $c^{y_{0:t-1}}_{2,t}$, $c^{y_{0:T}}_{3,T}$ and $c^{y_{0:T}}_{4,t}$ are defined in H4. Using H5, it is easy to prove that $\mathbb{E}[\kappa_1(y_{0:T})^2]$, $\mathbb{E}[\kappa_2(y_{0:T})^2]$, $\mathbb{E}[\kappa_3(y_{0:T})^2]$, and $\mathbb{E}[\kappa_4(y_{0:T})^2]$ are upper bounded by $cT^2$ for a constant $c$ that depends only on $\sigma_+$, $\sigma_-$ and $A$, and Theorem 3.1 follows.

# E   Additional proofs

**Proposition E.1.** *Assume that H1-3 hold. For all $\theta$, $\theta' \in \Theta$, and all $y_{0:T} \in \mathsf{Y}^{T+1}$,*

$$|\ell_T^{y_{0:T}}(\theta) - \ell_T^{y_{0:T}}(\theta')| \leq \kappa_1(y_{0:T})\|\theta - \theta'\|_2 \,,$$

*where*

$$\kappa_1(y_{0:T}) = \frac{\sigma_+\eta_+(G^{y_0})}{\sigma_-c_-(y_0)} + \sum_{t=1}^{T} \frac{\sigma_+}{\sigma_-c_-(y_t)} \left\{ c_+(y_t)L_{t-1}(y_{0:t-1}) + \frac{\eta_+ \otimes \mu(M \cdot \bar{g}^{y_{t-1}} \bar{\otimes} g^{y_t})}{\sigma_-c_-(y_{t-1})} + \eta_+(G^{y_t}) \right\} \,,$$

*with $M \cdot \bar{g}^{y_{t-1}} \otimes \bar{g}^{y_t}(x, x') = M(x, x')\bar{g}^{y_{t-1}}(x)\bar{g}^{y_t}(x')$, where $L_{t-1}$ is defined in Lemma F.2.*

*Proof.* For all $\theta$, $\theta' \in \Theta$, and all $y_{0:T} \in \mathsf{Y}^{T+1}$, with the convention $p_\theta(y_0|y_{-1}) = p_\theta(y_0)$,

$$\ell_T^{y_{0:T}}(\theta) - \ell_T^{y_{0:T}}(\theta') = \sum_{t=0}^{T} (\log p_\theta(y_t|y_{0:t-1}) - \log p_{\theta'}(y_t|y_{0:t-1})) \,.$$

For all $t > 0$,

$$p_\theta(y_t|y_{0:t-1}) = \int \Phi_{\theta,t-1}^{y_{0:t-1}}(\mathrm{d}x_{t-1})M_\theta(x_{t-1}, \mathrm{d}x_t)g_\theta^{y_t}(x_t) \,.$$

Note first that

$$p_\theta(y_t|y_{0:t-1}) \geq \sigma_-c_-(y_t) \,,$$

so that

$$|\ell_T^{y_{0:T}}(\theta) - \ell_T^{y_{0:T}}(\theta')| \leq \frac{|p_\theta(y_0) - p_{\theta'}(y_0)|}{\sigma_-c_-(y_0)} + \sum_{t=0}^{T} \frac{|p_\theta(y_t|y_{0:t-1}) - p_{\theta'}(y_t|y_{0:t-1})|}{\sigma_-c_-(y_t)} \,.$$

For $t = 0$, using that $p_\theta(y_0) = \int \chi(\mathrm{d}x_0)g_\theta^{y_0}(x_0)$, Assumptions H1 and H3 yield

$$|p_\theta(y_0) - p_{\theta'}(y_0)| \leq \sigma_+\eta_+(G^{y_0})\|\theta - \theta'\|_2 \,.$$

In addition,

$$p_\theta(y_t|y_{0:t-1}) - p_{\theta'}(y_t|y_{0:t-1}) = \int \left( \Phi_{\theta,t-1}^{y_{0:t-1}}(\mathrm{d}x_{t-1}) - \Phi_{\theta',t-1}^{y_{0:t-1}}(\mathrm{d}x_{t-1}) \right) M_\theta(x_{t-1}, \mathrm{d}x_t)g_\theta^{y_t}(x_t)$$

$$+ \int \Phi_{\theta',t-1}^{y_{0:t-1}}(\mathrm{d}x_{t-1}) \left( M_\theta(x_{t-1}, \mathrm{d}x_t) - M_{\theta'}(x_{t-1}, \mathrm{d}x_t) \right) g_\theta^{y_t}(x_t) + \int \Phi_{\theta',t-1}^{y_{0:t-1}}(\mathrm{d}x_{t-1})M_{\theta'}(x_{t-1}, \mathrm{d}x_t) \left( g_\theta^{y_t}(x_t) - g_{\theta'}^{y_t}(x_t) \right) \,.$$

Using Lemma F.1, Assumptions H1 and H3, we get

$$|p_\theta(y_t|y_{0:t-1}) - p_{\theta'}(y_t|y_{0:t-1})| \leq \left\{ \sigma_+c_+(y_t) \left\| \Phi_{\theta,t-1}^{y_{0:t-1}} - \Phi_{\theta',t-1}^{y_{0:t-1}} \right\|_{\mathrm{tv}} + \right.$$

$$\left. \frac{\sigma_+}{\sigma_-c_-(y_{t-1})} \int \eta_+ \otimes \mu(\mathrm{d}x\mathrm{d}x')(M(x, x')\bar{g}^{y_{t-1}}(x)\bar{g}^{y_t}(x')) + \sigma_+\eta_+(G^{y_t}) \right\} \|\theta - \theta'\|_2 \,.$$

The proof is completed by using Lemma F.2. $\qquad\square$

**Proposition E.2.** *Assume that H1-4 hold. Then,*

$$\Delta_1(\theta, \varphi, \varphi', y_{0:T}) \leq \kappa_2(y_{0:T})\|\varphi - \varphi'\|_2 \,,$$

*where*

$$\Delta_1(\theta, \varphi, \varphi', y_{0:T}) = \left| \mathbb{E}_{q_{\varphi,0:T}^{y_{0:T}}} \left[ \log \frac{q_{\varphi,0:T}^{y_{0:T}}(X_{0:T})}{\phi_{\theta,0:T|T}^{y_{0:T}}(X_{0:T})} \right] - \mathbb{E}_{q_{\varphi',0:T}^{y_{0:T}}} \left[ \log \frac{q_{\varphi,0:T}^{y_{0:T}}(X_{0:T})}{\phi_{\theta,0:T|T}^{y_{0:T}}(X_{0:T})} \right] \right| \,,$$

*with $\rho(y_{0:T}) = 1 - \vartheta_-^{y_{0:T}}$ and*

$$\kappa_2(y_{0:T}) = (\vartheta_+^{y_{0:T}})^3 \sum_{t=1}^{T} \upsilon_t^{y_{0:T}} \sum_{s=t-1}^{T} \lambda_+ \otimes \lambda_+(K_{s|s+1}^{y_{0:T}})\rho(y_{0:T})^{s-t} \,.$$

*Proof.* For all $\varphi, \varphi' \in \Phi$, $0 \le t \le T-1$, define

$$\tilde{q}^{y_{0:T}}_{\varphi,\varphi',t|T}(x_{0:T}) = q^{y_{0:T}}_{\varphi,T}(x_T) \prod_{u=T}^{t+1} q^{y_{0:T}}_{\varphi,u-1|u}(x_u, x_{u-1}) \prod_{u=t}^{1} q^{y_{0:T}}_{\varphi',u-1|u}(x_u, x_{u-1})$$

$$- q^{y_{0:T}}_{\varphi,T}(x_T) \prod_{u=T}^{t+2} q^{y_{0:T}}_{\varphi,u-1|u}(x_u, x_{u-1}) \prod_{u=t+1}^{1} q^{y_{0:T}}_{\varphi',u-1|u}(x_u, x_{u-1})$$

with the convention $\prod_{u=T}^{T+1} q^{y_{0:T}}_{\varphi,u-1|u}(x_u, x_{u-1}) = 1$ and $\prod_{u=0}^{1} q^{y_{0:T}}_{\varphi',u-1|u}(x_u, x_{u-1}) = 1$, and for $t = T$,

$$\tilde{q}^{y_{0:T}}_{\varphi,\varphi',T|T}(x_{0:T}) = q^{y_{0:T}}_{\varphi,T}(x_T) \prod_{u=T}^{1} q^{y_{0:T}}_{\varphi,u-1|u}(x_u, x_{u-1}) - q^{y_{0:T}}_{\varphi',T}(x_T) \prod_{u=T}^{1} q^{y_{0:T}}_{\varphi',u-1|u}(x_u, x_{u-1}) \ .$$

Therefore,

$$\Delta_1(\theta, \varphi, \varphi', y_{0:T}) = \left| \sum_{t=1}^{T} \mathbb{E}_{q^{y_{0:T}}_{\varphi,0:T}}\left[ h^{y_{0:T}}_{t,\theta,\varphi}(X_{t-1}, X_t) \right] - \mathbb{E}_{q^{y_{0:T}}_{\varphi',0:T}}\left[ h^{y_{0:T}}_{t,\theta,\varphi}(X_{t-1}, X_t) \right] \right| \ ,$$

$$= \left| \sum_{t=1}^{T} \sum_{s=0}^{T} \mathbb{E}_{\tilde{q}^{y_{0:T}}_{\varphi,\varphi',s|T}}\left[ h^{y_{0:T}}_{t,\theta,\varphi}(X_{t-1}, X_t) \right] \right| \ ,$$

where $h^{y_{0:T}}_{t,\theta,\varphi}$, $1 \le t \le T$, are defined in equation 4. Note first that if $t > s+1$, then $\mathbb{E}_{\tilde{q}^{y_{0:T}}_{\varphi,\varphi',s|T}}\left[ h^{y_{0:T}}_{t,\theta,\varphi}(X_{t-1}, X_t) \right] = 0$ so that

$$\Delta_1(\theta, \varphi, \varphi', y_{0:T}) = \left| \sum_{t=1}^{T} \sum_{s=t-1}^{T} \mathbb{E}_{\tilde{q}^{y_{0:T}}_{\varphi,\varphi',s|T}}\left[ h^{y_{0:T}}_{t,\theta,\varphi}(X_{t-1}, X_t) \right] \right| \ .$$

For all $t \le s+1$, write for all measurable set $A$,

$$\mu^{y_{0:T}}_{\varphi,s}(A) = \int \mathbb{1}_A(x_s) q^{y_{0:T}}_{\varphi,T}(x_T) \mu(\mathrm{d}x_T) \prod_{u=T}^{s+1} q^{y_{0:T}}_{\varphi,u-1|u}(x_u, x_{u-1}) \mu(\mathrm{d}x_{u-1}) \ ,$$

$$\tilde{\mu}^{y_{0:T}}_{\varphi,\varphi',s}(A) = \int \mathbb{1}_A(x_s) q^{y_{0:T}}_{\varphi,T}(x_T) \mu(\mathrm{d}x_T) \prod_{u=T}^{s+2} q^{y_{0:T}}_{\varphi,u-1|u}(x_u, x_{u-1}) \mu(\mathrm{d}x_{u-1}) q^{y_{0:T}}_{\varphi',s|s+1}(x_{s+1}, x_s) \mu(\mathrm{d}x_s) \ .$$

Therefore,

$$\mathbb{E}_{\tilde{q}^{y_{0:T}}_{\varphi,\varphi',s|T}}\left[ h^{y_{0:T}}_{t,\theta,\varphi}(X_{t-1}, X_t) \right] = \left( \mu^{y_{0:T}}_{\varphi,s} - \tilde{\mu}^{y_{0:T}}_{\varphi,\varphi',s} \right) \left\{ \prod_{u=s}^{t+1} Q^{y_{0:T}}_{\varphi',u-1|u} \right\} Q^{y_{0:T}}_{\varphi',t-1|t} h^{y_{0:T}}_{t,\theta,\varphi} \ .$$

Using H1, the backward variational kernels satisfy a Doeblin condition, see (Douc et al., 2014, Section 6.1.3), so that

$$\mathbb{E}_{\tilde{q}^{y_{0:T}}_{\varphi,\varphi',s|T}}\left[ h^{y_{0:T}}_{t,\theta,\varphi}(X_{t-1}, X_t) \right] \le \frac{1}{2} \|\mu^{y_{0:T}}_{\varphi,s} - \tilde{\mu}^{y_{0:T}}_{\varphi,\varphi',s}\|_{\mathrm{tv}} \rho(y_{0:T})^{s-t} \mathrm{osc}\left( Q^{y_{0:T}}_{\varphi',t-1|t} h^{y_{0:T}}_{t,\theta,\varphi} \right) \ ,$$

where for all measurable functions $f$, $\mathrm{osc}(f) = \sup_{x,x' \in \mathsf{X}} |f(x) - f(x')|$. By H1 and H4,

$$\mathrm{osc}\left( Q^{y_{0:T}}_{\varphi',t-1|t} h^{y_{0:T}}_{t,\theta,\varphi} \right) \le 2 \left\| \int q^{y_{0:T}}_{\varphi',t-1|t}(\cdot, x_{t-1}) h^{y_{0:T}}_{t,\theta,\varphi}(x_{t-1}, \cdot) \mu(\mathrm{d}x_{t-1}) \right\|_{\infty} \ ,$$

$$\le 2\vartheta^{y_{0:T}}_{+} \left\| \int \left| h^{y_{0:T}}_{t,\theta,\varphi}(x_{t-1}, \cdot) \right| \lambda_{+}(\mathrm{d}x_{t-1}) \right\|_{\infty} \ ,$$

$$\le 2\vartheta^{y_{0:T}}_{+} \upsilon^{y_{0:T}}_{t} \ .$$

Noting that by H3,

$$\|\mu_{\varphi,s}^{y_{0:T}} - \tilde{\mu}_{\varphi,\varphi',s}^{y_{0:T}}\|_{\mathrm{tv}} \leq Q_{\varphi,T}^{y_{0:T}} \prod_{s=T}^{t+1} Q_{\varphi,s-1|s}^{y_{0:T}} K_{s|s+1}^{y_{0:T}} \|\varphi - \varphi'\|_2 \leq (\vartheta_+^{y_{0:T}})^2 \lambda_+ \otimes \lambda_+(K_{s|s+1}^{y_{0:T}}) \|\varphi - \varphi'\|_2 \ ,$$

concludes the proof. $\qquad\square$

**Proposition E.3.** *Assume that H1-4 hold. Then,*

$$\Delta_2(\theta, \theta', \varphi, \varphi', y_{0:T}) \leq \kappa_3(y_{0:T}) \|\varphi - \varphi'\|_2 + \kappa_4(y_{0:T}) \|\theta - \theta'\|_2 \ ,$$

*where*

$$\Delta_2(\theta, \theta', \varphi, \varphi', y_{0:T}) = \left| \mathbb{E}_{q_{\varphi',0:T}^{y_{0:T}}} \left[ \log \frac{q_{\varphi,0:T}^{y_{0:T}}(X_{0:T})}{\phi_{\theta,0:T|T}^{y_{0:T}}(X_{0:T})} \right] - \mathbb{E}_{q_{\varphi',0:T}^{y_{0:T}}} \left[ \log \frac{q_{\varphi',0:T}^{y_{0:T}}(X_{0:T})}{\phi_{\theta',0:T|T}^{y_{0:T}}(X_{0:T})} \right] \right| \ ,$$

*with*

$$\kappa_3(y_{0:T}) = \vartheta_+^{y_{0:T}} \left( \vartheta_+^{y_{0:T}} \sum_{t=1}^{T} c_{1,t}^{y_{0:T}} + c_{3,T}^{y_{0:T}} \right) \quad \text{and} \quad \kappa_4(y_{0:T}) = \vartheta_+^{y_{0:T}} \left( \vartheta_+^{y_{0:T}} \sum_{t=1}^{T} c_{2,t}^{y_{0:t-1}} + c_{4,t}^{y_{0:T}} \right) \ ,$$

*and where* $c_{1,t}^{y_{0:T}}$, $c_{2,t}^{y_{0:t-1}}$, $c_{3,T}^{y_{0:T}}$ *and* $c_{4,t}^{y_{0:T}}$ *are defined in H4.*

*Proof.* By definition,

$$\Delta_2(\theta, \theta', \varphi, \varphi', y_{0:T}) = \left| \mathbb{E}_{q_{\varphi',0:T}^{y_{0:T}}} \left[ \log \frac{q_{\varphi,0:T}^{y_{0:T}}(X_{0:T})}{\phi_{\theta,0:T|T}^{y_{0:T}}(X_{0:T})} \right] - \mathbb{E}_{q_{\varphi',0:T}^{y_{0:T}}} \left[ \log \frac{q_{\varphi',0:T}^{y_{0:T}}(X_{0:T})}{\phi_{\theta',0:T|T}^{y_{0:T}}(X_{0:T})} \right] \right| \ ,$$

$$\leq \mathbb{E}_{q_{\varphi',0:T}^{y_{0:T}}} \left[ \left| \log \frac{q_{\varphi,0:T}^{y_{0:T}}(X_{0:T})}{\phi_{\theta,0:T|T}^{y_{0:T}}(X_{0:T})} - \log \frac{q_{\varphi',0:T}^{y_{0:T}}(X_{0:T})}{\phi_{\theta',0:T|T}^{y_{0:T}}(X_{0:T})} \right| \right] \ ,$$

$$\leq \sum_{t=1}^{T} \mathbb{E}_{q_{\varphi',0:T}^{y_{0:T}}} \left[ \left| h_{t,\theta,\varphi}^{y_{0:T}}(X_{t-1}, X_t) - h_{t,\theta',\varphi'}^{y_{0:T}}(X_{t-1}, X_t) \right| \right] \ ,$$

where $h_{t,\theta,\varphi}^{y_{0:T}}$, $1 \leq t \leq T$, are defined in equation 4. For $t < T$ and all $x_{t-1}, x_t \in \mathsf{X}$,

$$\left| h_{t,\theta,\varphi}^{y_{0:T}}(x_{t-1}, x_t) - h_{t,\theta',\varphi'}^{y_{0:T}}(x_{t-1}, x_t) \right| \leq \left| \log q_{\varphi,t-1|t}^{y_{0:T}}(x_t, x_{t-1}) - \log q_{\varphi',t-1|t}^{y_{0:T}}(x_t, x_{t-1}) \right|$$

$$+ \left| \log b_{\theta,t-1|t}^{y_{0:t-1}}(x_t, x_{t-1}) - \log b_{\theta',t-1|t}^{y_{0:t-1}}(x_t, x_{t-1}) \right| \ .$$

Using H1 and H4,

$$\mathbb{E}_{q_{\varphi',0:T}^{y_{0:T}}} \left[ \left| \log q_{\varphi,t-1|t}^{y_{0:T}}(x_t, x_{t-1}) - \log q_{\varphi',t-1|t}^{y_{0:T}}(x_t, x_{t-1}) \right| \right]$$

$$\leq (\vartheta_+^{y_{0:T}})^2 \int \lambda_+ \otimes \lambda_+(\mathrm{d}x\mathrm{d}x') \left| \log q_{\varphi,t-1|t}^{y_{0:T}}(x, x') - \log q_{\varphi',t-1|t}^{y_{0:T}}(x, x') \right| \ ,$$

$$\leq (\vartheta_+^{y_{0:T}})^2 c_{1,t}^{y_{0:T}} \|\varphi - \varphi'\|_2 \ .$$

Similarly,

$$\mathbb{E}_{q_{\varphi',0:T}^{y_{0:T}}} \left[ \left| \log b_{\theta,t-1|t}^{y_{0:t-1}}(x_t, x_{t-1}) - \log b_{\theta',t-1|t}^{y_{0:t-1}}(x_t, x_{t-1}) \right| \right]$$

$$\leq (\vartheta_+^{y_{0:T}})^2 \int \lambda_+ \otimes \lambda_+(\mathrm{d}x\mathrm{d}x') \left| \log b_{\theta,t-1|t}^{y_{0:t-1}}(x, x') - \log b_{\theta',t-1|t}^{y_{0:t-1}}(x, x') \right| \ ,$$

$$\leq (\vartheta_+^{y_{0:T}})^2 c_{2,t}^{y_{0:t-1}} \|\theta - \theta'\|_2 \ .$$

For $t = T$, it remains to bound $\mathbb{E}_{q_{\varphi',0:T}^{y_{0:T}}}[|\log q_{\varphi,T}^{y_{0:T}}(X_T) - \log q_{\varphi',T}^{y_{0:T}}(X_T)| + |\log \phi_{\theta,T}^{y_{0:T}}(X_T) - \log \phi_{\theta',T}^{y_{0:T}}(X_T)|]$, which is straightforward by using H1 and H4. $\qquad\square$

**Proposition E.4.** *Assume that H1-2 and H6 hold. Then, there exists $c > 0$ such that ,*

$$\left\| \sup_{\theta \in \Theta} \left| \log L_T^{Y_{0:T}}(\theta) \right| \right\|_{\psi_{\alpha_*}} \leq cT .$$

*Proof.* For all $\theta \in \Theta$, and all $y_{0:T} \in \mathsf{Y}^{T+1}$, with the convention $p_\theta(y_0|y_{-1}) = p_\theta(y_0)$,

$$\log L_T^{y_{0:T}}(\theta) = \ell_T^{y_{0:T}}(\theta) = \sum_{t=0}^{T} \log p_\theta(y_t|y_{0:t-1}) .$$

As $p_\theta(y_0) = \int \chi(\mathrm{d}x_0) g_\theta^{y_0}(x_0)$, by H1-2, $\sigma_- c_-(y_0) \leq p_\theta(y_0) \leq \sigma_+ c_+(y_0)$. For all $t > 0$,

$$p_\theta(y_t|y_{0:t-1}) = \int \Phi_{\theta,t-1}^{y_{0:t-1}}(\mathrm{d}x_{t-1}) M_\theta(x_{t-1}, \mathrm{d}x_t) g_\theta^{y_t}(x_t) ,$$

so that by H1-2 $\sigma_- c_-(y_t) \leq p_\theta(y_t|y_{0:t-1}) \leq \sigma_+ c_+(y_t)$. Using the second point in H6 and the triangular inequality concludes the proof. $\qquad\square$

**Proposition E.5.** *Assume that H1 and H6 hold. Then, there exists $B > 0$ such that*

$$\left\| \sup_{\theta \in \Theta, \varphi \in \Phi, \chi} \left| \mathrm{KL}\left( Q_{\varphi,0:T}^{Y_{0:T}} \middle\| \phi_{\theta,T}^{Y_{0:T}} \right) \right| \right\|_{\psi_{\alpha_*}} \leq BT ,$$

*Proof.* For all $\theta \in \Theta$, $\varphi \in \Phi$, $y_{0:T} \in \mathsf{Y}^{T+1}$,

$$\mathrm{KL}\left( Q_{\varphi,0:T}^{y_{0:T}} \middle\| \phi_{\theta,T}^{y_{0:T}} \right) = \mathbb{E}_{q_{\varphi,0:T}^{y_{0:T}}} \left[ \log \frac{q_{\varphi,0:T}^{y_{0:T}}(X_{0:T})}{\phi_{\theta,0:T|T}^{y_{0:T}}(X_{0:T})} \right] = \sum_{t=1}^{T} \mathbb{E}_{q_{\varphi,0:T}^{y_{0:T}}} \left[ h_{t,\theta,\varphi}^{y_{0:T}}(X_{t-1}, X_t) \right] ,$$

where $h_{t,\theta,\varphi}^{y_{0:T}}$, $1 \leq t \leq T$, are defined in equation 4. By H1, for all $1 \leq t \leq T$,

$$\left| \mathbb{E}_{q_{\varphi,0:T}^{y_{0:T}}} \left[ h_{t,\theta,\varphi}^{y_{0:T}}(X_{t-1}, X_t) \right] \right| \leq (\vartheta_+^{y_{0:T}})^2 \lambda_+ \otimes \lambda_+ \left( \left| h_{t,\theta,\varphi}^{y_{0:T}} \right| \right) ,$$

which concludes the proof by H6. $\qquad\square$

## F  Technical results

**Lemma F.1.** *Assume that H1 and H2 hold. For all $\theta \in \Theta$, all $t \geq 0$, all $y_{0:t} \in \mathsf{Y}^{T+1}$, positive measurable function $h$,*

$$\frac{\sigma_- \eta_-(g_\theta^{y_t} h)}{\sigma_+ c_+(y_t)} \leq \Phi_{\theta,t}^{y_{0:t}}(h) \leq \frac{\sigma_+ \eta_+(g_\theta^{y_t} h)}{\sigma_- c_-(y_t)} .$$

*Proof.* At time 0, we have $\Phi_{\theta,0}^{y_0}(\mathrm{d}x_0) \propto \chi(\mathrm{d}x_0) g_\theta^{y_0}(x_0)$ so that by H1-2,

$$\frac{\sigma_- \eta_-(g_\theta^{y_0} h)}{\sigma_+ c_+(y_0)} \leq \Phi_{\theta,0}^{y_0}(h) \leq \frac{\sigma_+ \eta_+(g_\theta^{y_0} h)}{\sigma_- c_-(y_0)} .$$

Similarly,

$$\Phi_{\theta,t}^{y_{0:t}}(\mathrm{d}x_t) \propto g_\theta^{y_t}(x_t) \int \Phi_{\theta,t-1}^{y_{0:t-1}}(\mathrm{d}x_{t-1}) M_\theta(x_{t-1}, \mathrm{d}x_t) ,$$

so that by H1 and H2,

$$\frac{\sigma_- \eta_-(g_\theta^{y_t} h)}{\sigma_+ c_+(y_t)} \leq \Phi_{\theta,t}^{y_{0:t}}(h) \leq \frac{\sigma_+ \eta_+(g_\theta^{y_t} h)}{\sigma_- c_-(y_t)} .$$

$\qquad\square$

**Lemma F.2.** *Assume that H1, H2 and H3 hold. Then, for all $\theta$, $\theta' \in \Theta$, $t \geq 1$,*

$$\left\| \Phi_{\theta,t}^{y_{0:t}} - \Phi_{\theta',t}^{y_{0:t}} \right\|_{\mathrm{tv}} \leq L_t(y_{0:t}) \|\theta - \theta'\|_2 \,,$$

*where*

$$L_t(y_{0:t}) = \frac{4\sigma_+^2}{\sigma_-^2} \sum_{s=0}^{t} \varepsilon^{t-s} \frac{1}{c_-(y_s)} \left\{ \frac{1}{\sigma_- c_-(y_{s-1})} \eta_+ \otimes \mu \left(\bar{g}^{y_{s-1}} \otimes \bar{g}^{y_s} \cdot M\right) + \eta_+(G^{y_s}) \right\} \,,$$

*with $\varepsilon = 1 - \sigma_-/\sigma_+$.*

*Proof.* The proof follows the same lines as the proof of (De Castro et al., 2017, Proposition 2.1), which was in the setting of a discrete state space. For $t > 0$, note that $\Phi_{\theta,t}^{y_{0:t}}(\mathrm{d}x_t) = g_\theta^{y_t}(x_t) \int \Phi_{\theta,t-1}^{y_{0:t-1}}(\mathrm{d}x_{t-1}) M_\theta(x_{t-1}, \mathrm{d}x_t)/c_{\theta,t}(y_{0:t})$ where $c_{\theta,t}(y_{0:t}) = \int g_\theta^{y_t}(x_t)\Phi_{\theta,t-1}^{y_{0:t-1}}(\mathrm{d}x_{t-1}) M_\theta(x_{t-1}, \mathrm{d}x_t)$. Consider the forward kernel at time $t$ defined, for all $\theta \in \Theta$, all $y_t \in \mathsf{Y}$, $x \in \mathbb{R}^d$, and probability measure $\gamma$ by

$$F_{\theta,t}^{y_t} \gamma(x) = \frac{\int m_\theta(x', x) g_\theta^{y_t}(x) \gamma(\mathrm{d}x')}{\int m_\theta(x', x'') g_\theta^{y_t}(x'') \gamma(\mathrm{d}x')\mu(\mathrm{d}x'')} \,.$$

Therefore, $\Phi_{\theta,t}^{y_{0:t}} = F_{\theta,t}^{y_t}\Phi_{\theta,t-1}^{y_{0:t-1}}$ and for all $\theta, \theta' \in \Theta$,

$$\Phi_{\theta,t}^{y_{0:t}} - \Phi_{\theta',t}^{y_{0:t}} = F_{\theta,t}^{y_t}\Phi_{\theta,t-1}^{y_{0:t-1}} - F_{\theta',t}^{y_t}\Phi_{\theta',t-1}^{y_{0:t-1}} \,,$$

$$= \sum_{s=0}^{t-1} \Delta_{t,s}(y_{s:t}) + F_{\theta,t}^{y_t}\Phi_{\theta',t-1}^{y_{0:t-1}} - F_{\theta',t}^{y_t}\Phi_{\theta',t-1}^{y_{0:t-1}} \,,$$

where

$$\Delta_{t,s}(y_{s:t}) = F_{\theta,t}^{y_t} \cdots F_{\theta,s+1}^{y_{s+1}} F_{\theta,s}^{y_s} \Phi_{\theta',s-1}^{y_{0:s-1}} - F_{\theta,t}^{y_t} \cdots F_{\theta,s+1}^{y_{s+1}} \Phi_{\theta',s}^{y_{0:s}}$$

with the convention $F_{\theta,0}^{y_0}\Phi_{\theta',-1}^{y_{-1}} = \Phi_{\theta,0}^{y_0}$. Consider also the backward function $\beta_{s|t}^{y_{s+1:t}}$ and the forward smoothing kernel $F_{s|t,\theta}^{y_{s:t}}$ defined by

$$\beta_{\theta,s|t}^{y_{s+1:t}}(x_s) = \int M_\theta(x_s, \mathrm{d}x_{s+1}) g_\theta^{y_{s+1}}(x_{s+1}) \cdots M_\theta(x_{t-1}, \mathrm{d}x_t) g_\theta^{y_t}(x_t) \,,$$

$$F_{\theta,s|t}^{y_{s:t}}(x_{s-1}, x_s) = \frac{\beta_{s|t}^{y_{s+1:t}}(x_s) m_\theta(x_{s-1}, x_s) g_\theta^{y_s}(x_s)}{\int \beta_{s|t}^{y_{s+1:t}}(x) M_\theta(x_{s-1}, \mathrm{d}x) g_\theta^{y_s}(x)} \,.$$

Following for instance (Cappé et al., 2005, Chapter 4), we can write for all probability measure $\gamma$,

$$F_{\theta,t}^{y_t} \cdots F_{\theta,s+1}^{y_{s+1}} \gamma = \gamma_{\theta,s|t} F_{\theta,s+1|t}^{y_{s+1:t}} \cdots F_{\theta,t|t}^{y_t} \,,$$

where $\gamma_{\theta,s|t} \propto \beta_{\theta,s|t}^{y_{s+1:t}} \gamma$. Therefore,

$$\Phi_{\theta,t}^{y_{0:t}} - \Phi_{\theta',t}^{y_{0:t}} = \sum_{s=0}^{t-1} \left( \gamma_{\theta,\theta',s|t} F_{\theta,s+1|t}^{y_t} \cdots F_{\theta,t|t}^{y_{s+1}} - \tilde{\gamma}_{\theta,\theta',s|t} F_{\theta,s+1|t}^{y_t} \cdots F_{\theta,t|t}^{y_{s+1}} \right) + F_{\theta,t}^{y_t}\Phi_{\theta',t-1}^{y_{0:t-1}} - F_{\theta',t}^{y_t}\Phi_{\theta',t-1}^{y_{0:t-1}} \,,$$

where $\gamma_{\theta,\theta',s|t} \propto \beta_{\theta,s|t}^{y_{s+1:t}} F_{\theta,s}^{y_s} \Phi_{\theta',s-1}^{y_{0:s-1}}$ and $\tilde{\gamma}_{\theta,\theta',s|t} \propto \beta_{\theta,s|t}^{y_{s+1:t}} \Phi_{\theta',s}^{y_{0:s}}$. Note that by H1, for all measurable sets $A$,

$$F_{\theta,s|t}^{y_{s:t}}(x_{s-1}, A) \geq \frac{\sigma_-}{\sigma_+} \frac{\int \eta_-(\mathrm{d}x_s)\mathbb{1}_A(x)\beta_{s|t}^{y_{s+1:t}}(x)g_\theta^{y_s}(x)}{\int \eta_+(\mathrm{d}x)\beta_{s|t}^{y_{s+1:t}}(x)g_\theta^{y_s}(x)} \,,$$

so that

$$\left\| \gamma_{\theta,\theta',s|t} F_{\theta,s+1|t}^{y_t} \cdots F_{\theta,t|t}^{y_{s+1}} - \tilde{\gamma}_{\theta,\theta',s|t} F_{\theta,s+1|t}^{y_t} \cdots F_{\theta,t|t}^{y_{s+1}} \right\|_{\mathrm{tv}} \leq \epsilon^{t-s} \left\| \gamma_{\theta,\theta',s|t} - \tilde{\gamma}_{\theta,\theta',s|t} \right\|_{\mathrm{tv}} \,,$$

with $\epsilon = 1 - \sigma_-/\sigma_+$. This yields

$$\left\| \Phi_{\theta,t}^{y_{0:t}} - \Phi_{\theta',t}^{y_{0:t}} \right\|_{\mathrm{tv}} \leq \sum_{s=0}^{t-1} \epsilon^{t-s} \left\| \gamma_{\theta,\theta',s|t} - \tilde{\gamma}_{\theta,\theta',s|t} \right\|_{\mathrm{tv}} + \left\| F_{\theta,t}^{y_t} \Phi_{\theta',t-1}^{y_{0:t-1}} - F_{\theta',t}^{y_t} \Phi_{\theta',t-1}^{y_{0:t-1}} \right\|_{\mathrm{tv}} .$$

For all bounded measurable functions $h$,

$$\left| \gamma_{\theta,\theta',s|t}(h) - \tilde{\gamma}_{\theta,\theta',s|t}(h) \right| = \left| \frac{\int \beta_{\theta,s|t}^{y_{s+1:t}}(x_s) F_{\theta,s}^{y_s} \Phi_{\theta',s-1}^{y_{0:s-1}}(x_s) h(x_s) \mu(\mathrm{d}x_s)}{\int \beta_{\theta,s|t}^{y_{s+1:t}}(x_s) F_{\theta,s}^{y_s} \Phi_{\theta',s-1}^{y_{0:s-1}}(x_s) \mu(\mathrm{d}x_s)} - \frac{\int \beta_{\theta,s|t}^{y_{s+1:t}}(x_s) \Phi_{\theta',s}^{y_{0:s}}(x_s) h(x_s) \mu(\mathrm{d}x_s)}{\int \beta_{\theta,s|t}^{y_{s+1:t}}(x_s) \Phi_{\theta',s}^{y_{0:s}}(x_s) \mu(\mathrm{d}x_s)} \right| ,$$

$$\leq \delta_{\theta,\theta',1}^{y_{0:t}}(h) + \delta_{\theta,\theta',2}^{y_{0:t}}(h) ,$$

where

$$\delta_{\theta,\theta',1}^{y_{0:t}}(h) = \frac{\int \beta_{\theta,s|t}^{y_{s+1:t}}(x_s) \left| F_{\theta,s}^{y_s} \Phi_{\theta',s-1}^{y_{0:s-1}}(x_s) - F_{\theta',s}^{y_s} \Phi_{\theta',s-1}^{y_{0:s-1}}(x_s) \right| h(x_s) \mu(\mathrm{d}x_s)}{\int \beta_{\theta,s|t}^{y_{s+1:t}}(x_s) F_{\theta,s}^{y_s} \Phi_{\theta',s-1}^{y_{0:s-1}}(x_s) \mu(\mathrm{d}x_s)} ,$$

$$\delta_{\theta,\theta',2}^{y_{0:t}}(h) = \frac{\int \beta_{\theta,s|t}^{y_{s+1:t}}(x_s) \Phi_{\theta',s}^{y_{0:s}}(x_s) h(x_s) \mu(\mathrm{d}x_s)}{\int \beta_{\theta,s|t}^{y_{s+1:t}}(x_s) \Phi_{\theta',s}^{y_{0:s}}(x_s) \mu(\mathrm{d}x_s)} \frac{\int \beta_{\theta,s|t}^{y_{s+1:t}}(x_s) \left| F_{\theta,s}^{y_s} \Phi_{\theta',s-1}^{y_{0:s-1}}(x_s) - F_{\theta',s}^{y_s} \phi_{\theta',s-1}^{y_{0:s-1}}(x_s) \right| \mu(\mathrm{d}x_s)}{\int \beta_{\theta,s|t}^{y_{s+1:t}}(x_s) F_{\theta,s}^{y_s} \Phi_{\theta',s-1}^{y_{0:s-1}}(x_s) \mu(\mathrm{d}x_s)} .$$

Note that for all $x_s \in \mathsf{X}$, by H1,

$$\sigma_- \int \eta_-(\mathrm{d}x_{s+1}) g_\theta^{y_{s+1}}(x_{s+1}) \cdots m_\theta(x_{t-1}, x_t) g_\theta^{y_t}(x_t) \mu(\mathrm{d}x_{s+2:t}) \leq \beta_{\theta,s|t}^{y_{s+1:t}}(x_s)$$

$$\leq \sigma_+ \int \eta_+(\mathrm{d}x_{s+1}) g_\theta^{y_{s+1}}(x_{s+1}) \cdots m_\theta(x_{t-1}, x_t) g_\theta^{y_t}(x_t) \mu(\mathrm{d}x_{s+2:t}) ,$$

so that

$$\delta_{\theta,\theta',1}^{y_{0:t}}(h) + \delta_{\theta,\theta',2}^{y_{0:t}}(h) \leq 2\|h\|_\infty \|F_{\theta,s}^{y_s} \Phi_{\theta',s-1}^{y_{0:s-1}} - F_{\theta',s}^{y_s} \Phi_{\theta',s-1}^{y_{0:s-1}}\|_{\mathrm{tv}} \frac{\|\beta_{\theta,s|t}^{y_{s+1:t}}\|_\infty}{\inf_{x \in \mathsf{X}} \beta_{\theta,s|t}^{y_{s+1:t}}(x_s)}$$

$$\leq 2\frac{\sigma_+}{\sigma_-} \|h\|_\infty \|F_{\theta,s}^{y_s} \Phi_{\theta',s-1}^{y_{0:s-1}} - F_{\theta',s}^{y_s} \Phi_{\theta',s-1}^{y_{0:s-1}}\|_{\mathrm{tv}} .$$

For all bounded measurable function $h$,

$$\left| F_{\theta,s}^{y_s} \Phi_{\theta',s-1}^{y_{0:s-1}} h - F_{\theta',s}^{y_s} \Phi_{\theta',s-1}^{y_{0:s-1}} h \right| \leq R_1 + R_2 ,$$

where

$$R_1 = \left| \frac{\int \left( m_\theta(x', x) g_\theta^{y_s}(x) - m_{\theta'}(x', x) g_\theta^{y_s}(x) \right) \Phi_{\theta',s-1}^{y_{0:s-1}}(\mathrm{d}x') h(x) \mu(\mathrm{d}x)}{\int m_\theta(x', x'') g_\theta^{y_s}(x'') \Phi_{\theta',s-1}^{y_{0:s-1}}(\mathrm{d}x') \mu(\mathrm{d}x'')} \right| ,$$

$$R_2 = \left| \frac{\int m_{\theta'}(x', x'') g_{\theta'}^{y_s}(x'') \Phi_{\theta',s-1}^{y_{0:s-1}}(\mathrm{d}x') h(x'') \mu(\mathrm{d}x'')}{\int m_{\theta'}(x', x'') g_{\theta'}^{y_s}(x'') \Phi_{\theta',s-1}^{y_{0:s-1}}(\mathrm{d}x') \mu(\mathrm{d}x'')} \right| \cdot \left| \frac{\int \left( m_\theta(x', x'') g_\theta^{y_s}(x'') - m_{\theta'}(x', x'') g_{\theta'}^{y_s}(x'') \right) \Phi_{\theta',s-1}^{y_{0:s-1}}(\mathrm{d}x') \mu(\mathrm{d}x'')}{\int m_\theta(x', x'') g_\theta^{y_s}(x'') \Phi_{\theta',s-1}^{y_{0:s-1}}(\mathrm{d}x') \mu(\mathrm{d}x'')} \right| .$$

By H1-3 and Lemma F.1,

$$R_1 \leq \frac{\sigma_+}{\sigma_- c_-(y_s)} \left\{ \frac{1}{\sigma_- c_-(y_{s-1})} \eta_+ \otimes \mu \left( \bar{g}^{y_{s-1}} \otimes \bar{g}^{y_s} \cdot M \right) + \eta_+(G^{y_s}) \right\} \|\theta - \theta'\|_2 \|h\|_\infty$$

The same upper bound can be obtained for $R_2$ as

$$\left| \frac{\int m_{\theta'}(x', x'') g_{\theta'}^{y_s}(x'') \Phi_{\theta',s-1}^{y_{0:s-1}}(\mathrm{d}x') h(x'') \mu(\mathrm{d}x'')}{\int m_{\theta'}(x', x'') g_{\theta'}^{y_s}(x'') \Phi_{\theta',s-1}^{y_{0:s-1}}(\mathrm{d}x') \mu(\mathrm{d}x'')} \right| \leq \|h\|_\infty .$$

This yields

$$\|F_{\theta,s}^{y_s} \Phi_{\theta',s-1}^{y_{0:s-1}} - F_{\theta',s}^{y_s} \Phi_{\theta',s-1}^{y_{0:s-1}}\|_{\mathrm{tv}} \leq \frac{2\sigma_+}{\sigma_- c_-(y_s)} \left\{ \frac{1}{\sigma_- c_-(y_{s-1})} \eta_+ \otimes \mu \left( \bar{g}^{y_{s-1}} \otimes \bar{g}^{y_s} \cdot M \right) + \eta_+(G^{y_s}) \right\} \|\theta - \theta'\|_2 ,$$

which concludes the proof. $\qquad\square$

## G   Checking assumptions

In this section, we provide additional assumptions on the state space model and on the variational family to support that our assumptions can be verified.

**A1** There exist constants $0 < \sigma_- < \sigma_+ < \infty$ such that for all $x \in \mathsf{X}$,

$$\sigma_- \leq \zeta(x) \leq \sigma_+$$

and for all $\theta \in \Theta$, $x, x' \in \mathsf{X}$,

$$\sigma_- \leq m_\theta(x, x') \leq \sigma_+ \ .$$

For all $y_{0:T} \in \mathsf{Y}^{T+1}$, there exist $\vartheta_-^{y_{0:T}} > 0$ and $\vartheta_+^{y_{0:T}} > 0$ such that for all $\varphi \in \Phi$, $t \geq 0$, all $x, x' \in \mathsf{X}$,

$$\vartheta_-^{y_{0:T}} \leq q_{\varphi,t|t+1}^{y_{0:T}}(x, x') \leq \vartheta_+^{y_{0:T}} \ .$$

In addition, for all $\varphi \in \Phi$, all $y_{0:T} \in \mathsf{Y}^{T+1}$, and all $x \in \mathsf{X}$,

$$\vartheta_-^{y_{0:T}} \leq q_{\varphi,T}^{y_{0:T}}(x) \leq \vartheta_+^{y_{0:T}} \ .$$

Assumption A1 is known as a strong-mixing assumption and allows to verify H1. It is classical to obtain quantitative bounds on approximation of joint smoothing distributions, see for instance Olsson et al. (2008); Gloaguen et al. (2022). It typically requires the state space $\mathsf{X}$ to be compact. In settings where the bacwkard variartional kernels are Gaussian and obtained with neural networks which are uniformly bounded with respect to the time index and the observations, $\vartheta_+^{y_{0:T}}$ and $\vartheta_-^{y_{0:T}}$ do not depend on the observations.

**A2** For all $y \in \mathsf{Y}$, $\inf_{\theta \in \Theta} \int g_\theta^y(x)\mu(\mathrm{d}x) = c_-(y) > 0$ and $\sup_{\theta \in \Theta} \int g_\theta^y(x)\mu(\mathrm{d}x) = c_+(y) < \infty$.

Lemma G.1, Lemma G.2 and Proposition G.3 allow to obtain explicit constants in H4. We prove that the functions $h_{t,\theta,\varphi}^{y_{0:T}}$ are upper-bounded explicitly, and that $\phi_{\theta,t}^{y_{0:t}}$ and $b_{\theta,t-1|t}^{y_{0:t-1}}$ are lower and upper-bounded explicitly, in particular with respect to the observation sequence.

When the observation space is compact we can also obtain a uniform control with respect to the observations of these quantities which is crucial to check H5 and H6.

**Lemma G.1.** *Assume that A1 and A2 hold. For all $\theta \in \Theta$, all $t \geq 0$, all $y_{0:t}$, $x_t$,*

$$\frac{\sigma_- g_\theta^{y_t}(x_t)}{\sigma_+ c_+(y_t)} \leq \phi_{\theta,t}^{y_{0:t}}(x_t) \leq \frac{\sigma_+ g_\theta^{y_t}(x_t)}{\sigma_- c_-(y_t)} \ .$$

*Proof.* At time 0, we have $\phi_{\theta,0}^{y_0}(x_0) \propto \zeta(x_0)g_\theta^{y_0}(x_0)$ so that by A1-2,

$$\frac{\sigma_- g_\theta^{y_0}(x_0)}{\sigma_+ c_+(y_0)} \leq \phi_{\theta,0}^{y_0}(x_0) \leq \frac{\sigma_+ g_\theta^{y_0}(x_0)}{\sigma_- c_-(y_0)} \ .$$

Similarly,

$$\phi_{\theta,t}^{y_{0:t}}(x_t) \propto g_\theta^{y_t}(x_t) \int \Phi_{\theta,t-1}^{y_{0:t-1}}(\mathrm{d}x_{t-1})m_\theta(x_{t-1}, x_t)\mu(\mathrm{d}x_t) \,,$$

so that by A1 and A2,

$$\frac{\sigma_- g_\theta^{y_t}(x_t)}{\sigma_+ c_+(y_t)} \leq \phi_{\theta,t}^{y_{0:t}}(x_t) \leq \frac{\sigma_+ \eta g_\theta^{y_t}(x_t)}{\sigma_- c_-(y_t)} \ .$$

$\square$

**Lemma G.2.** *Assume that A1 and A2 hold. For all $\theta$, all $1 \leq t \leq T$, all $y_{0:T}$, $x_{t-1}$, $x_t$,*

$$\frac{\sigma_-^2 g_\theta^{y_{t-1}}(x_{t-1})}{\sigma_+^2 c_+(y_{t-1})} \leq b_{\theta,t-1|t}^{y_{0:t-1}}(x_t, x_{t-1}) \leq \frac{\sigma_+^2 g_\theta^{y_{t-1}}(x_{t-1})}{\sigma_-^2 c_-(y_{t-1})}$$

*and for $1 \leq t \leq T-1$,*

$$\|h_{t,\theta,\varphi}^{y_{0:T}}\|_\infty \leq |\log \vartheta_-(y_{0:T})| \vee |\log \vartheta_+(y_{0:T})|$$
$$+ \sup_{x_{t-1} \in \mathsf{X}} \left| \log \frac{\sigma_-^2 c_-(y_{t-1}) \underline{g}^{y_{t-1}}(x_{t-1})}{\sigma_+^2 c_+(y_{t-1})} \right| \vee \left| \log \frac{\sigma_+^2 c_+(y_{t-1}) \bar{g}^{y_{t-1}}(x_{t-1})}{\sigma_-^2 c_-(y_{t-1})} \right|$$

*and*

$$\|h_{T,\theta,\varphi}^{y_{0:T}}\|_\infty \leq |\log 2\vartheta_-^{y_{0:T}}| \vee |\log 2\vartheta_+^{y_{0:T}}| + \sup_{x_T \in \mathsf{X}} \left| \log \frac{\sigma_- \underline{g}^{y_T}(x_T)}{\sigma_+ c_+(y_T)} \right| \vee \left| \log \frac{\sigma_+ \bar{g}^{y_T}(x_T)}{\sigma_- c_-(y_T)} \right|$$
$$+ \sup_{x_{T-1} \in \mathsf{X}} \left| \log \frac{\sigma_-^2 c_-(y_{T-1}) \underline{g}^{y_{T-1}}(x_{T-1})}{\sigma_+^2 c_+(y_{T-1})} \right| \vee \left| \log \frac{\sigma_+^2 c_+(y_{T-1}) \bar{g}^{y_{T-1}}(x_{T-1})}{\sigma_-^2 c_-(y_{T-1})} \right|,$$

*where $h_{t,\theta,\varphi}$, $1 \leq t \leq T$, are defined in equation 4.*

*Proof.* By Lemma F.1,

$$\frac{\sigma_-^2 g_\theta^{y_{t-1}}(x_{t-1})}{\sigma_+ c_+(y_{t-1})} \leq \phi_{\theta,t-1}^{y_{0:t-1}}(x_{t-1}) m_\theta(x_{t-1}, x_t) \leq \frac{\sigma_+^2 g_\theta^{y_{t-1}}(x_{t-1})}{\sigma_- c_-(y_{t-1})} \ .$$

Since

$$b_{\theta,t-1|t}^{y_{0:t-1}}(x_t, x_{t-1}) = \frac{\phi_{\theta,t-1}^{y_{0:t-1}}(x_{t-1}) m_\theta(x_{t-1}, x_t)}{\int \phi_{\theta,t-1}^{y_{0:t-1}}(x_{t-1}) m_\theta(x_{t-1}, x_t) \mu(\mathrm{d}x_{t-1})}$$

we get

$$\frac{\sigma_-^2 c_-(y_{t-1}) g_\theta^{y_{t-1}}(x_{t-1})}{\sigma_+^2 c_+(y_{t-1})} \leq b_{\theta,t-1|t}^{y_{0:t-1}}(x_t, x_{t-1}) \leq \frac{\sigma_+^2 c_+(y_{t-1}) g_\theta^{y_{t-1}}(x_{t-1})}{\sigma_-^2 c_-(y_{t-1})}.$$

Now by equation 4, for $1 \leq t \leq T-1$, $h_{t,\theta,\varphi}^{y_{0:T}}(x_{t-1}, x_t) = \log q_{\varphi,t-1|t}^{y_{0:T}}(x_t, x_{t-1}) - \log b_{\theta,t-1|t}^{y_{0:t-1}}(x_t, x_{t-1})$ so that

$$\left| h_{t,\theta,\varphi}^{y_{0:T}}(x_{t-1}, x_t) \right| \leq |\log \vartheta_-(y_{0:T})| \vee |\log \vartheta_+(y_{0:T})|$$
$$+ \left| \log \frac{\sigma_-^2 c_-(y_{t-1}) g_\theta^{y_{t-1}}(x_{t-1})}{\sigma_+^2 c_+(y_{t-1})} \right| \vee \left| \log \frac{\sigma_+^2 c_+(y_{t-1}) g_\theta^{y_{t-1}}(x_{t-1})}{\sigma_-^2 c_-(y_{t-1})} \right|,$$

which concludes the proof. In addition, using that

$$h_{T,\theta,\varphi}^{y_{0:T}}(x_{T-1}, x_T) = \log q_{\varphi,T-1|T}^{y_{0:T}}(x_T, x_{T-1}) - \log b_{\theta,T-1|T}^{y_{0:T-1}}(x_T, x_{T-1}) + \log q_{\varphi,T}^{y_{0:T}}(x_T) - \log \phi_{\theta,T}^{y_{0:T}}(x_T)$$

yields

$$\left| h_{T,\theta,\varphi}^{y_{0:T}}(x_{T-1}, x_t) \right| \leq |\log 2\vartheta_-(y_{0:T})| \vee |\log 2\vartheta_+(y_{0:T})| + \left| \log \frac{\sigma_- g_\theta^{y_T}(x_T)}{\sigma_+ c_+(y_T)} \right| \vee \left| \log \frac{\sigma_+ g_\theta^{y_T}(x_T)}{\sigma_- c_-(y_T)} \right|$$
$$+ \left| \log \frac{\sigma_-^2 c_-(y_{T-1}) g_\theta^{y_{T-1}}(x_{T-1})}{\sigma_+^2 c_+(y_{T-1})} \right| \vee \left| \log \frac{\sigma_+^2 c_+(y_{T-1}) g_\theta^{y_{T-1}}(x_{T-1})}{\sigma_-^2 c_-(y_{T-1})} \right| \ .$$

$\square$

**Proposition G.3.** *Assume that A1, A2 and H3 hold. Then H4 holds. More precisely, for all $y_{0:T} \in \mathsf{Y}^{T+1}$ and all $0 \le t \le T$,*

$$\sup_{\theta \in \Theta, \varphi \in \Phi} \left\| \int \lambda(\mathrm{d}x) \left| h_{t,\theta,\varphi}^{y_{0:T}}(x, \cdot) \right| \right\|_\infty = \upsilon_t^{y_{0:T}} < \infty \ ,$$

*where $\upsilon_t^{y_{0:T}} = \sup_{\theta \in \Theta, \varphi \in \Phi} \| h_{t,\theta,\varphi}^{y_{0:T}} \|_\infty$ is given in Lemma G.2. For all $\theta, \theta' \in \Theta$, $\varphi, \varphi' \in \Phi$, $1 \le t \le T$,*

$$\int \lambda \otimes \lambda(\mathrm{d}x\mathrm{d}x') \left| \log q_{\varphi,t-1|t}^{y_{0:T}}(x, x') - \log q_{\varphi',t-1|t}^{y_{0:T}}(x, x') \right| \le c_{1,t}^{y_{0:T}} \| \varphi - \varphi' \|_2 \ ,$$

$$\int \lambda \otimes \lambda(\mathrm{d}x\mathrm{d}x') \left| \log b_{\theta,t-1|t}^{y_{0:t-1}}(x, x') - \log b_{\theta',t-1|t}^{y_{0:t-1}}(x, x') \right| \le c_{2,t}^{y_{0:t-1}} \| \theta - \theta' \|_2 \ ,$$

$$\int \lambda(\mathrm{d}x) \left| \log q_{\varphi,T}^{y_{0:T}}(x) - \log q_{\varphi',T}^{y_{0:T}}(x) \right| \le c_{3,T}^{y_{0:T}} \| \varphi - \varphi' \|_2 \ ,$$

$$\int \lambda(\mathrm{d}x) \left| \log \phi_{\theta,T}^{y_{0:T}}(x) - \log \phi_{\theta',T}^{y_{0:T}}(x) \right| \le c_{4,T}^{y_{0:T}} \| \theta - \theta' \|_2 \ ,$$

*where $c_{1,t}^{y_{0:T}} = (\vartheta_-^{y_{0:T}})^{-1} \lambda \otimes \lambda(K_{t-1|t}^{y_{0:T}})$, $c_{2,t}^{y_{0:t-1}} = 2\sigma_+ L_{t-1}(y_{0:t-1})/(\sigma_- \inf_{x \in \mathsf{X}} \underline{g}^{y_{t-1}}(x))$, $c_{3,t}^{y_{0:T}} = (\vartheta_-^{y_{0:T}})^{-1} \lambda(K_T^{y_{0:T}})$, and $c_{4,T}^{y_{0:T}} = 2\sigma_+ c_+(y_T) L_T(y_{0:T})/(\sigma_- \inf_{x \in \mathsf{X}} \underline{g}^{y_T}(x))$.*

*Proof.* For all $\varphi, \varphi' \in \Phi$, $1 \le t \le T$,

$$\left| \log q_{\varphi,t-1|t}^{y_{0:T}}(x, x') - \log q_{\varphi',t-1|t}^{y_{0:T}}(x, x') \right| \le \frac{\left| q_{\varphi,t-1|t}^{y_{0:T}}(x, x') - q_{\varphi',t-1|t}^{y_{0:T}}(x, x') \right|}{\left| q_{\varphi,t-1|t}^{y_{0:T}}(x, x') \wedge q_{\varphi',t-1|t}^{y_{0:T}}(x, x') \right|} \ ,$$

so that by A1 and H3,

$$\left| \log q_{\varphi,t-1|t}^{y_{0:T}}(x, x') - \log q_{\varphi',t-1|t}^{y_{0:T}}(x, x') \right| \le (\vartheta_-^{y_{0:T}})^{-1} K_{t-1|t}^{y_{0:T}}(x', x) \| \varphi - \varphi' \| \ ,$$

an we can choose $c_{1,t}^{y_{0:T}} = (\vartheta_-^{y_{0:T}})^{-1} \lambda \otimes \lambda(K_{t-1|t}^{y_{0:T}})$. Similarly, for all $\varphi, \varphi' \in \Phi$,

$$\left| \log q_{\varphi,T}^{y_{0:T}}(x) - \log q_{\varphi',T}^{y_{0:T}}(x) \right| \le \frac{\left| q_{\varphi,T}^{y_{0:T}}(x) - q_{\varphi',T}^{y_{0:T}}(x) \right|}{\left| q_{\varphi,T}^{y_{0:T}}(x) \wedge q_{\varphi',T}^{y_{0:T}}(x) \right|} \ ,$$

so that by A1 and H3,

$$\left| \log q_{\varphi,T}^{y_{0:T}}(x) - \log q_{\varphi',T}^{y_{0:T}}(x) \right| \le (\vartheta_-^{y_{0:T}})^{-1} K_T^{y_{0:T}}(x) \| \varphi - \varphi' \| \ ,$$

and we can choose $c_{3,t}^{y_{0:T}} = (\vartheta_-^{y_{0:T}})^{-1} \lambda(K_T^{y_{0:T}})$. For all $\theta, \theta' \in \Theta$, $1 \le t \le T$,

$$\left| \log b_{\theta,t-1|t}^{y_{0:t-1}}(x, x') - \log b_{\theta',t-1|t}^{y_{0:t-1}}(x, x') \right| \le \frac{\left| b_{\theta,t-1|t}^{y_{0:t-1}}(x, x') - b_{\theta',t-1|t}^{y_{0:t-1}}(x, x') \right|}{\left| b_{\theta,t-1|t}^{y_{0:t-1}}(x, x') \wedge b_{\theta',t-1|t}^{y_{0:t-1}}(x, x') \right|} \ .$$

By Lemma G.2,

$$\left| \log b_{\theta,t-1|t}^{y_{0:t-1}}(x, x') - \log b_{\theta',t-1|t}^{y_{0:t-1}}(x, x') \right| \le \frac{\sigma_+^2 c_+(y_{t-1})}{\sigma_-^2 \underline{g}^{y_{t-1}}(x_{t-1})} \left| b_{\theta,t-1|t}^{y_{0:t-1}}(x, x') - b_{\theta',t-1|t}^{y_{0:t-1}}(x, x') \right| \ .$$

Then, noting that $b_{\theta,t-1|t}^{y_{0:t-1}}(x, x') = \phi_{\theta,t-1}^{y_{0:t-1}}(x') m_\theta(x', x)/c_\theta(x)$ where $c_\theta(x) = \int \phi_{\theta,t-1}^{y_{0:t-1}}(x') m_\theta(x', x) \mu(\mathrm{d}x')$, write

$$\left| b_{\theta,t-1|t}^{y_{0:t-1}}(x, x') - b_{\theta',t-1|t}^{y_{0:t-1}}(x, x') \right| \le \left| \frac{(\phi_{\theta,t-1}^{y_{0:t-1}}(x') - \phi_{\theta',t-1}^{y_{0:t-1}}(x')) m_\theta(x', x)}{c_\theta(x)} \right|$$

$$+ \left| \frac{\phi_{\theta',t-1}^{y_{0:t-1}}(x')(m_\theta(x', x) - m_{\theta'}(x', x))}{c_\theta(x)} \right| + \left| \frac{\phi_{\theta',t-1}^{y_{0:t-1}}(x') m_{\theta'}(x', x)}{c_{\theta'}(x)} \right| \left| \frac{c_{\theta'}(x) - c_\theta(x)}{c_\theta(x)} \right|$$

By A1,

$$\int \lambda \otimes \lambda(\mathrm{d}x\mathrm{d}x') \left| \frac{(\phi_{\theta,t-1}^{y_{0:t-1}}(x') - \phi_{\theta',t-1}^{y_{0:t-1}}(x'))m_\theta(x',x)}{\underline{g}^{y_{t-1}}(x')c_\theta(x)} \right| \leq 2\frac{\sigma_+}{\sigma_- \inf_{x\in\mathsf{X}}\underline{g}^{y_{t-1}}(x)} \left\| \Phi_{\theta,t-1}^{y_{0:t-1}} - \Phi_{\theta',t-1}^{y_{0:t-1}} \right\|_{\mathrm{tv}} ,$$

and by Lemma G.2, we can choose $c_{2,t}^{y_{0:t-1}} = 2\sigma_+ L_{t-1}(y_{0:t-1})/(\sigma_- \inf_{x\in\mathsf{X}}\underline{g}^{y_{t-1}}(x))$. For all $\theta, \theta' \in \Theta$,

$$\left| \log\phi_{\theta,T}^{y_{0:T}}(x) - \log\phi_{\theta',T}^{y_{0:T}}(x) \right| \leq \frac{\left| \phi_{\theta,T}^{y_{0:T}}(x) - \phi_{\theta',T}^{y_{0:T}}(x) \right|}{\left| \phi_{\theta,T}^{y_{0:T}}(x) \wedge \phi_{\theta',T}^{y_{0:T}}(x) \right|} ,$$

By Lemma G.1,

$$\left| \log\phi_{\theta,T}^{y_{0:T}}(x) - \log\phi_{\theta',T}^{y_{0:T}}(x) \right| \leq \frac{\sigma_+ c_+(y_T)}{\sigma_- \underline{g}^{y_T}(x)} \left| \phi_{\theta,T}^{y_{0:T}}(x) - \phi_{\theta',T}^{y_{0:T}}(x) \right| .$$

Therefore,

$$\int \lambda(\mathrm{d}x) \left| \log\phi_{\theta,T}^{y_{0:T}}(x) - \log\phi_{\theta',T}^{y_{0:T}}(x) \right| \leq 2\frac{\sigma_+ c_+(y_T)}{\sigma_- \inf_{x\in\mathsf{X}}\underline{g}^{y_T}(x)} \left\| \Phi_{\theta,T}^{y_{0:T}} - \Phi_{\theta',T}^{y_{0:T}} \right\|_{\mathrm{tv}} ,$$

and by Lemma G.2, we can choose $c_{4,T}^{y_{0:T}} = 2\sigma_+ c_+(y_T)L_T(y_{0:T})/(\sigma_- \inf_{x\in\mathsf{X}}\underline{g}^{y_T}(x))$. $\qquad\square$

If the observation space is compact, under standard regularity assumptions, all upper bounds can be obtained uniformly with respect to the observations. Therefore, H5 holds as soon as the integrals under $\mu$, $\eta_+ \otimes \mu$ and $\lambda_+ \otimes \lambda_+$ are finite.

