# OpenReview forum: "Variational excess risk bound  for general state space models"
_TMLR — Accepted by TMLR_

### Review · Reviewer_qS78 · 2024-02-06

**Summary Of Contributions:**

This paper applies the result of Tang & Yang 2021 to the case of variational inference within state space models. The result bounds the sum of the KL divergence between the data distribution over observations and the model distribution for the observations plus the KL divergence between the variational posterior and true posterior. The bound is based on the error introduced when optimizing the ELBO based on a finite number of observations.
The authors find that the bound has a similar form as Tang & Yang but depends on the cube of the length of the sequence.
The paper then considers common parameterizations for state space models and considers whether the assumptions of the theorem are satisfied for these parameterizations.

**Audience:**

Yes

**Broader Impact Concerns:**

I have no concerns of the ethical impacts of the work.

**Claims And Evidence:**

Yes

**Requested Changes:**

None of these concerns are critical for my recommendation of acceptance.

1) I assume zeta refers to the initial distribution but this was not defined.
2) It is not ideal to have m refer to both the transition distribution and the single sample ELBO, it would be good to change notation for one of these quantities.
3) In the proof of theorem 3.1, Condition A of the theorem is referred to but is referred to but is not actually explicitly stated until the Appendix. I think at least a reference to the appendix is needed here or putting condition A in the main text.
4) P^Y is only mentioned in the intro but I should be given a proper definition as it is used later on without any explicit definition given.
5) It should be made clear what role the backward factorization plays in the derivation as this is mentioned during the motivation of the work. Is the theorem not possible to derive with a forward factorization?
6) Some discussion of the derived result would be nice, e.g. the bound depends on the cube of the length of the observed sequence. However, in the work of Chagneux et al. (2022), the error seems to depend linearly on the length of the sequence. Is there any insight to be gained from considering why there is this discrepancy?

**Strengths And Weaknesses:**

Strengths:
- The paper offers a novel application of the VAE excess risk bound to state space models offering some theoretical backing to these techniques.
- The paper thoroughly examines the assumptions required to derive the theorem and discuss their applicability in common state space modeling scenarios.
- The claims made by the paper clearly match the derived technical results and contributions.

Weaknesses (requested change number in brackets):
- The writing could be improved by making some notations clearer and introducing definitions where they are needed. (1,2,3,4)
- There could be more discussion around the specific implications of the result in the state space modeling setting. (6)
- It is not clear how the use of the backward factorization links to the stated result. (5)

---

> ### Author Response · Authors · 2024-04-07
>
> We thank the referee for the careful and insightful review of our manuscript. We provide below an answer associated with each requested change.
>
> **1**. zeta was/is defined in the third line of Section 2 as the density of the initial distribution.
>
> **2**. Yes you are right. We changed the notation for the single sample ELBO.
>
> **3**. Thank you. We chose to refer to the appendix.
>
> **4**.  You are right, these notations are not required in the introduction. We decided to introduce them in Section 2 with all other important notations of the paper.
>
> **5**.  In state space models, the true posterior distribution of the latent states given the observations admits a backward Markovian decomposition. Therefore, this factorization allows to introduce a variational family which fits the data structure which is not the case of mean-field approximations. Other factorizations could be used for the variational family but this backward factorization is appealing for several reasons: (i) the variational laws admits the same decomposition as the target laws, (ii) the Markovian decomposition is useful to derive theoretical controls based on forgetting properties of Markov chains and (iii) many practical parameterizations can be used for the kernels using recurrent neural networks. This is now more clearly discussed in the paper.
>
> **6**.  This is a very good question. In Chagneux et al. (2022) the authors control the variational posterior error when the objective is to compute expectations of additive functionals. Under similar assumptions as the strong mixing assumptions of our paper, they obtain a linear growth in $T$. In our setting we target a more challenging objective i.e. obtaining excess risk bounds and a full control on the divergence between the true data distribution and the estimated one (Corollary 3.2). Indeed, as an intermediate step, the proof requires similar bounds as the one in Chagneux et al. (2022). We added a remark after our main result to discuss this. We do not claim that our upper bounds is optimal but it is not surprising that it is less sharp than the one of Chagneux et al. (2022).

---

> > ### Comment · Reviewer_qS78 · 2024-04-08
> > **Reply to authors**
> >
> > Thank you for your responses. They all make sense to me however I would just like to clarify point regarding 5). When you say the mean field approximation couldn't fit the data structure, do you mean a variational posterior of the form Q(x_1 | y_1) Q(x_2 | y_2) Q(x_3 | y_3)... ? As I can see that couldn't fit the data structure. But I am more interested in the reasoning behind the backward factorization as opposed to a forward factorization, both of which can fit the true data structure, i.e. using a factorization of the form Q(x_1 | y_{1:T}) Q(x_2 | x_1, y_{2:T}) Q(x_3 | x_2, y_{3:T})... It would seem to me that this forward factorization still admits the same decomposition as the target law, and you could parameterize this using a recurrent neural network by having a forward backward approach where the backward encodes all the observations and the forward generates your variational distribution parameters. Is the forward factorization somehow less amenable to using the forgetting properties of Markov chains as you state in point ii) ?

---

> > > ### Author Response · Authors · 2024-04-08
> > >
> > > Thank you for this answer.
> > >
> > > You are right that it is possible to introduce a forward Markovian decomposition which fits the decomposition of the true model. We overlooked this decomposition in our answer. So it is true that we could introduce a variational family with this decomposition. In this setting, the forward Markov kernel at time $t$ involves the prior Markov kernel and the conditional distribution of $y_{t:T}$ given $x_t$. This decomposition shares some similarities with the two-filter decomposition in state space models and could be investigated further.
> > >
> > > In practice we believe that the backward decomposition is more appealing since it requires to approximate the backward kernel which depends on the filtering distributions which enjoy forgetting properties. The forward recursion involves smoothing quantities (for instance the initial distribution is the smoothing distribution of $x_1$ given $y_{1:T}$) which are more challenging to estimate. However, it is true that a full comparison discussing in particular deep architectures for both approaches should be performed to conclude that an approach should be preferred to the other.
> > >
> > > Nonetheless, we thank the reviewer for this important remark and will add this clarification in the revised paper.

---

### Review · Reviewer_BwgR · 2024-03-01

**Summary Of Contributions:**

This paper tackles the problem of providing oracle inequalities for variational autoencoders (VAEs). Notably, the paper offers an oracle inequality under strong mixing assumptions for both the state space model and variational distribution.

**Audience:**

Yes

**Broader Impact Concerns:**

None.

**Claims And Evidence:**

Yes

**Requested Changes:**

Notation: it seems that $\Phi^{y_{0:t}}$ is used both for the distribution of $X_{0:T}$ given $Y_{0:T} = y_{0:T}$ (when $t=T$) and for the distribution of $X_t$ given $Y_{0:t} = y_{0:t}$, which is an overload of notation; could you please clarify?

The manuscript would greatly benefit from a simplification of notation and a more intuitive presentation of the assumptions. Instead of tackling the general case first, is it possible to strip the model down to the simplest possible while retaining features of the problem (e.g., the functional autoregressive model discussed in the examples section) and present the assumptions and result tailored for this particular example before moving to the full results?

A section early in the paper providing context and foundational understanding of variational inference, especially for readers unfamiliar with the field, would be highly valuable. For example, my understanding is that the $Q$ kernels are taken in applications to be parametrized by neural networks. Could you clarify how $(\widehat \theta_{n,T}, \widehat\varphi_{n,T})$ is computed in practice? If the optimization is intractable, please clarify that this paper addresses the idealized setting in which the optimization can be carried out exactly.

Include a technical overview section that outlines the high-level ideas of the proofs, distinguishing novel contributions from established techniques. Currently my impression is that the analysis rests on standard methods.

The paper should discuss the expected scaling of constants involved in the assumptions, particularly in light of their potential exponential scaling with ambient dimension.

Typos:
- Pg. 2, "using a mean-field approximations" -> "using a mean-field approximation"
- Pg. 2, "given for all" -> "given for every"
- Pg. 3, "ELBO is closed" -> "ELBO is close"
- Pg. 4, "close to minimize" -> "close to minimizing"

**Strengths And Weaknesses:**

Strengths:

The paper provides a concrete mathematical foundation for learning VAEs, which is a problem of great practical interest. Due to the complexity of the setup and analysis, I was not able to verify the proofs, but they seem careful and sophisticated.

Weaknesses:

The paper suffers from excessive notational complexity and abstract presentation, which greatly hinders accessibility and broader comprehension among readers not deeply versed in variational inference. I would like to note that part of TMLR’s acceptance criteria is that the work should be of interest to TMLR’s audience, and a prerequisite for this is for the content of the paper to be understandable.

Despite the mathematical rigor, the practical applicability of the derived oracle inequalities is not immediately clear, particularly given the abundance of assumptions required for the analysis. For example, the analysis hides constants involving density ratio bounds, and such constants could be considered to scale exponentially in the dimension in many situations. The paper contains little discussion of how to interpret the bounds.

---

> ### Author Response · Authors · 2024-04-07
>
> We thank the referee for the careful and insightful review of our manuscript.
>
> **Notation**. This is indeed an overload of notation which is now clearly stated. We thank the reviewer for this remark.
>
> **Simplification of the presentation**. Following this remark and additional remarks by other reviewers, we simplified the presentation of our main results.
>
>  - The assumptions are presented briefly in Section 3 and technical details are postponed in the appendices to highlight the main results.
>  - We provide only a sketch of proof of Theorem 3.1 in the main paper to highlight the main steps and the novelties. All technicalities are postponed to the appendices.
> - In addition,  discussions on several models based on neural networks and possible implementations of the kernels are given in Section 2 to focus on specific settings where the proposed variational approximations can be used.
>
> **Variational inference**. We added in Section 2 a subsection to detail specific choices of forward and backward kernels. In Section 2.2 we added a discussion on the practical computation of the loss function and we detailed  how the optimization problem is usually solved in practice for unfamiliar readers.
>
> **High level ideas**. We postponed the complete proof of our main result to the appendix and highlighted the steps of the proofs in the main paper to focus on high level ideas. We thank the reviewer for this comment which improves the readability of the paper.
>
> **Scaling with dimension**. This is a very interesting question yet also challenging. In our assumptions, many constants depend on the dimension (in particular assumptions on the moments and mixing assumptions on the kernels). In this paper we decided to set the focus on the dependency on the number of samples and on $T$. A reason is that tracking the dependency on the dimension is challenging in state space models and as stated by the reviewer some constants may scale exponentially with $d$. In our opinion, tracking the dependency on $d$ for mixing constants is a contribution on its own for general state spaces without assuming too restrictive conditions on the model. We agree that this is important and we added a comment after our main results to highlight that this is a limitation of our work and that it should be the focus of future works.

---

### Review · Reviewer_78cA · 2024-03-25

**Summary Of Contributions:**

The paper studies VAEs for state space models with a backward factorization proposed by Campbell et al. (2021).
An oracle inequality for the risk in terms of the number of samples and the length of the observation sequences is derived under strong mixing assumptions and with independent sequence trajectories.
When data is generated by a state space model, an upper bound for the KL divergence between data and its estimator under the variational distribution is derived; the bound is in terms of how the backward coding kernels approximate the backward decoding kernels.
Gaussian backward kernels with dense and recurrent neural networks are used to show that these theoretical results hold in practice.

**Audience:**

Yes

**Claims And Evidence:**

Yes

**Requested Changes:**

The presentation can be made more accessible for non-theoretical audience.
- Perhaps adding a paragraph motivating (1) why/how these results are useful and (2) how they could guide algorithm design.
- Following about, a dedicated related work section to explain what kind of theoretical results are obtained by previous work in similar/related setups can be useful. Following this, missing gaps (like what this work focuses on) can be identified.

## Misc
- `\citep` should be used for the two references in the title
- I didn't know find where $M_\theta$ (in H1) is first defined.

**Strengths And Weaknesses:**

## Strengths
- The paper provides some theoretical guarantees for VAEs for state space models.
- The results are novel and seem to be sound.

## Weaknesses
- The draft is not very accessible to non-theoretical audience. It requires a good understand of the literature to understand and appreciate the paper.

---

> ### Author Response · Authors · 2024-04-07
>
> We thank the referee for the careful and insightful review of our manuscript.
>
> We agree that the paper is not easily accessible as it combines several frameworks including hidden Markov models, variational inference and M-estimation theory. We propose to extend the introduction to insist on the motivations and on the fact that obtaining theoretical results about VAEs for HMMs has crucial consequences as it provides insights on the role of several hyperparameters, in our case the number of observations and the number of samples. In addition, following your review we propose the following modifications.
>
> - A discussion on existing theoretical results related to our work to highlight the specific role of the number of samples and of observations in the introduction. We also insist on the fact that our results are the first excess risk bounds in a context of VAE for state space models.
> - The introduction of several examples in Section 2 to discuss choices of parameterization for the state space models and the variational family and references to other practical examples.
> - We postponed all technical assumptions in the appendices to highlight the main results and quantitative bounds with respect to $T$ and $n$ and to avoid too many technicalities in the main paper.

---

### Author Response · Authors · 2024-04-07

Dear reviewers, we thank you for your comments on the paper.
In addition to our answers, we uploaded a new version of the paper, we hope that this answers your concerns.
We have highlighted the essential modifications in blue.

---

### Comment · Action_Editor_JLic · 2024-06-18
**Missing acceptance date**

Hi

Please can you update the camera ready pdf to include the proper publication in the header?

Cheers

Tom

---

### Decision · Action_Editor_JLic · 2024-05-14

**Recommendation:** Accept as is

**Comment:**

After the revisions, all reviewers were in favor of the paper being accepted, an opinion that concur with myself.  The general consensus was that even though the entirely theoretical and somewhat inaccessible nature of the work will reduce how widespread the scope of its appeal will be, it is still sound and potentially useful work that will be of interest to some in the TMLR audience.

As there are no notable outstanding concerns, I recommend accepting the paper "as is".  However, I would encourage the authors to try and make some further edits to improve readability and notational overhead for the camera-ready if possible.

**Audience:**

While the reviewers and I had some concerns about the significance of the work (in particular with regard to its practical applicability and transferrable intuitions), all agreed that the presented theory would still be of interest to some in the TMLR audience.

All reviewers raised concerns about the clarity of the manuscript, particularly in terms of its heavy notation, lack of sufficient intuition, and general inaccessibility to non-technical audiences.  This has been improved somewhat in the updated manuscript and all reviewers agreed that the clarity is now sufficient to not prohibit publication.  I personally think there is still probably more that could be done to further improve the accessibility and readability of the paper, but I also appreciate that this can be challenging for work of this kind of technical nature.

**Claims And Evidence:**

All reviewers were happy with the technical soundness of the work and the appropriateness of the claims made.  The work is entirely theoretical and while it would have benefited from some experimental evaluation (e.g. to aid understanding of the derived bounds), the reviewers did not feel that this left unjustified claims.